# GST-UNet: A Neural Framework for Spatiotemporal Causal Inference with Time-Varying Confounding

**Miruna Oprescu**
Cornell University
amo78@cornell.edu

**David K Park**
Brookhaven National Laboratory
dpark1@bnl.gov

**Xihaier Luo**
Brookhaven National Laboratory
xluo@bnl.gov

**Shinjae Yoo**
Brookhaven National Laboratory
sjyoo@bnl.gov

**Nathan Kallus**
Cornell University & Netflix
kallus@cornell.edu

## Abstract

Estimating causal effects from spatiotemporal observational data is essential in public health, environmental science, and policy evaluation, where randomized experiments are often infeasible. Existing approaches, however, either rely on strong structural assumptions or fail to handle key challenges such as interference, spatial confounding, temporal carryover, and *time-varying confounding*—where covariates are influenced by past treatments and, in turn, affect future ones. We introduce the **GST-UNet** (**G**-computation **S**patio-**T**emporal **UNet**), a theoretically grounded neural framework that combines a U-Net-based spatiotemporal encoder with regression-based iterative G-computation to estimate location-specific potential outcomes under complex intervention sequences. GST-UNet explicitly adjusts for time-varying confounders and captures non-linear spatial and temporal dependencies, enabling valid causal inference from a *single* observed trajectory in data-scarce settings. We validate its effectiveness in synthetic experiments and in a real-world analysis of wildfire smoke exposure and respiratory hospitalizations during the 2018 California Camp Fire. Together, these results position GST-UNet as a **principled and ready-to-use framework** for spatiotemporal causal inference, advancing reliable estimation in policy-relevant and scientific domains.

## 1 Introduction

Environmental hazards, public health interventions, and socio-economic policies often require understanding complex cause-and-effect relationships across space and time [30, 34, 41]. For instance, evaluating the health impacts of air quality regulations requires assessing how interventions influence both immediate outcomes and downstream effects across regions. Such applications demand robust tools for estimating causal effects from observational spatiotemporal data.

However, causal inference in spatiotemporal settings poses unique challenges. Outcomes are influenced not only by local covariates and interventions but also by those of neighboring regions (spatial confounding and interference). Effects may persist and accumulate over time (temporal carryover), and covariates often evolve in response to past interventions while simultaneously affecting future ones (time-varying confounding). For example, air quality regulations are often implemented in reaction to recent pollution levels and hospitalizations, which themselves shape future exposures and health outcomes—creating feedback loops that violate standard independence assumptions. These complexities induce bias in naive estimators and are especially challenging in single-trajectory settings, where replication across units or time is infeasible.

39th Conference on Neural Information Processing Systems (NeurIPS 2025).

Existing approaches offer limited solutions: classical methods rely on rigid structural assumptions or user-defined exposure mappings, while recent neural models emphasize predictive accuracy over causal identification. Many assume independent time series or model only spatial correlations, leaving a gap in methods that can jointly address interference, temporal dependencies, and evolving confounding within a principled causal framework (see Section 2).

To bridge this gap, we introduce **GST-UNet** (**G**-computation **S**patio-**T**emporal **UNet**), a theoretically grounded neural framework for estimating location-specific potential outcomes in spatiotemporal settings with time-varying confounding. GST-UNet builds on formal identification and consistency results derived under a representation-based time-invariance assumption, showing how causal effects can be recovered from a single observed trajectory. We then instantiate this theory in a practical neural architecture: a U-Net encoder with ConvLSTM and attention modules coupled to an iterative G-computation procedure that performs recursive causal adjustment over time. To ensure stable estimation over long horizons, we design a curriculum-based training strategy that gradually refines recursive pseudo-outcomes, enabling effective learning even in data-scarce regimes. Unlike existing approaches, GST-UNet requires no user-specified structural models and can be directly deployed in real-world spatiotemporal applications.

Our contributions are threefold: (1) We develop the first unified framework that couples theoretical identification and consistency guarantees with an end-to-end neural implementation for spatiotemporal causal inference; (2) We demonstrate through controlled simulations that GST-UNet robustly handles interference, temporal carryover, and time-varying confounding; and (3) We illustrate its practical value via a real-world analysis of wildfire smoke exposure and respiratory hospitalizations during the 2018 California Camp Fire.

In summary, GST-UNet provides a **principled and ready-to-use framework** for causal inference from spatiotemporal data, combining formal guarantees with a flexible neural implementation. By abstracting away model-specific assumptions, GST-UNet makes spatiotemporal causal estimation both **accessible and reliable** for applied scientific and policy domains.

## 2 Related Work

We summarize the most relevant prior work here, with a more detailed discussion in Appendix A.

**Classical Spatiotemporal Causal Inference.** Early approaches (e.g., spatial econometrics [2], difference-in-differences [20], synthetic controls [4]) rely on strong assumptions such as parallel trends and no interference. More recent methods incorporate time-varying confounding using inverse propensity weighting (IPW) and marginal structural models [31, 49], but cannot address interference unless via user-specified exposure mappings or hyper-local assumptions [11, 44, 48]. As noted by Zhou et al. [49], the literature remains sparse, particularly in settings with rich feedback dynamics.

**Machine Learning for Spatiotemporal Modeling.** Deep learning models for prediction–e.g., CNNs and RNNs [40, 47], graph-based methods [25, 46], and video transformers [6, 27]–capture complex spatial-temporal patterns but do not incorporate causal adjustments, and thus cannot estimate counterfactuals or adjust for time-varying confounders.

**Time Series Causal Inference.** Causal methods for longitudinal data include marginal structural models [36], iterative G-computation [35], and recent ML-based extensions using recurrent networks, transformers, or meta-learners [7, 16, 18, 24, 28, 39]. However, these assume access to independent time series (*e.g.* across patients) and cannot model cross-unit interactions in spatiotemporal settings.

**Neural-Based Spatiotemporal Causal Inference.** Tec et al. [42] propose a UNet-based model that adjusts for non-local spatial confounding but focuses on static exposures and does not address interference or time-varying effects. Most similar to our work, [1] presents a climate-focused model that shares certain architectural similarities but emphasizes prediction rather than causal adjustment, leaving causal identification under time-varying confounding largely unaddressed.

**Positioning of Our Work.** Our work bridges these threads by uniting a theoretically grounded G-computation framework with a neural architecture for spatiotemporal data. Unlike prior time-series methods that assume independent units or spatial models that overlook confounding feedback, GST-UNet is the first end-to-end approach that (i) establishes identification and consistency under explicit assumptions for a *single* spatiotemporal trajectory, and (ii) implements this theory in a practical neural model capable of handling interference, spatial confounding, and time-varying dynamics.

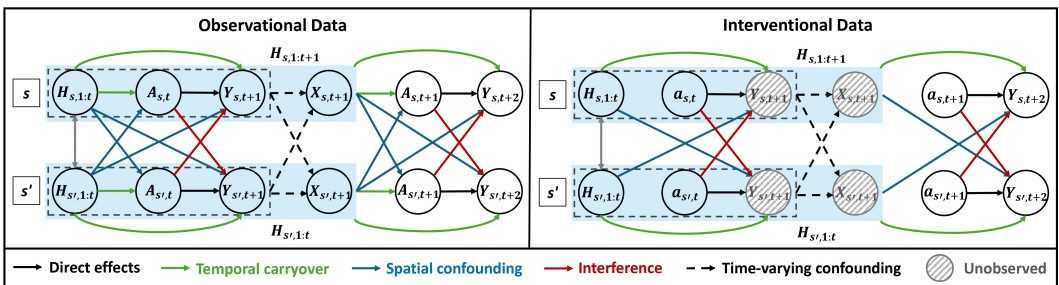

Figure 1: Observational data (left) versus interventional data (right) for a horizon $\tau = 2$ across multiple locations $(s, s')$. Under the intervention (right), treatments are set independently of confounders, and the full history is not observed for the entire horizon.

## 3 Problem Formulation

**Spatiotemporal Data.** We model observed data as random variables on a discrete spatial domain represented by an $N_X \times N_Y$ lattice: $\mathcal{S} = \{(i,j) \mid i \in [N_X], j \in [N_Y]\}$, where $[N] = \{1, \ldots, N\}$ denotes the index set. Time is indexed by $t \in [T]$. At each spatial location $s = (i, j)$ at time $t$, we observe a tuple $(\mathbf{X}_{s,t}, A_{s,t}, Y_{s,t})$, where $A_{s,t} \in \{0, 1\}$ represents a binary treatment (or intervention), $Y_{s,t} \in \mathbb{R}$ is a continuous outcome of interest, and $\mathbf{X}_{s,t} \in \mathbb{R}^{d_X}$ is a vector of time-varying covariates (*e.g.* local weather conditions, pollution levels, or socioeconomic indicators). Additionally, each location $s$ is associated with static features $V_s \in \mathbb{R}^{d_v}$ (*e.g.* geographical characteristics and socioeconomic indicators). While we focus on binary interventions for clarity, the methods generalize to more complex treatments. Conceptually, each variable forms a 3D spatiotemporal tensor of size $T \times N_X \times N_Y$, though in practice, observations may be incomplete. Missing data can be accommodated using masking techniques during downstream modeling.

To streamline notation, we use boldface symbols for random variables defined over the entire spatial domain. For $U \in \{X, A, Y\}$, let $\mathbf{U}_t$ denote its value at time $t$, and let $\mathbf{U}_{t:t+\tau} = (\mathbf{U}_t, \ldots, \mathbf{U}_{t+\tau})$ denote its value over a time interval. For a specific location $s$, we write $U_{s,t:t+\tau} = (U_{s,t}, \ldots, U_{s,t+\tau})$. The history up to time $t$ is denoted by $\mathbf{H}_{1:t} = (\mathbf{X}_{1:t}, \mathbf{A}_{1:t-1}, \mathbf{Y}_{1:t}, \mathbf{V})$ for the entire spatial domain and $H_{s,1:t} = (X_{s,1:t}, A_{s,1:t-1}, Y_{s,1:t}, V_s)$ for a specific location $s$. Specific instantiations of these random variables are denoted using lowercase letters (e.g., $u \in \{x, a, y, h\}$).

**Quantities of Interest.** Our primary goal is to estimate location-specific Conditional Average Potential Outcomes (CAPOs) for a sequence of future spatiotemporal interventions, conditioned on observed history. Our approach builds on Rubin's potential outcomes framework [35, 36, 38], which we extend to accommodate spatiotemporal settings. More concretely, we consider a future time horizon of length $\tau \geq 1$ and a predetermined interventional sequence $\mathbf{a}_{t:t+\tau-1}$ applied across the spatial domain starting at time $t$. Our goal is to estimate the potential outcomes at time $t + \tau$, denoted as $\mathbf{Y}_{t+\tau}[\mathbf{a}_{t:t+\tau-1}]$. In particular, we aim to compute:

$$\mathbb{E}[\mathbf{Y}_{t+\tau}[\mathbf{a}_{t:t+\tau-1}] \mid \mathbf{H}_{1:t} = \mathbf{h}_{1:t}] \tag{1}$$

which represents the CAPOs at time $t + \tau$ under the given treatment sequence. Given two different interventional sequences $\mathbf{a}_{t:t+\tau}$ and $\mathbf{a}'_{t:t+\tau}$, a related secondary goal is to estimate the location specific Conditional Average Treatment Effect (CATE), given by:

$$\mathbb{E}[\mathbf{Y}_{t+\tau}[\mathbf{a}_{t:t+\tau-1}] - \mathbf{Y}_{t+\tau}[\mathbf{a}'_{t:t+\tau-1}] \mid \mathbf{H}_{1:t} = \mathbf{h}_{1:t}]$$

Although we focus primarily on CAPOs, CATEs and other effect measures can be derived similarly.

**Prefix Data in a Single Spatiotemporal Chain.** The conditional expectations defining the CAPOs in Eq. (1) cannot be directly estimated from a single observed spatiotemporal realization, since the empirical averages would contain only one sample of each future outcome $\mathbf{Y}_{t+\tau}[\mathbf{a}_{t:t+\tau-1}]$. To obtain a workable regression-based estimator, we therefore reorganize the single observed trajectory into overlapping "prefixes" of varying lengths. For each $t \in \{1, \ldots, T - \tau\}$, we define

$$\mathbf{P}_t^\tau = (\mathbf{X}_{1:t+\tau}, \mathbf{A}_{1:t+\tau}, \mathbf{Y}_{1:t+\tau}, \mathbf{V}),$$

which represents the observed history up to time $t + \tau$ along with all covariates, treatments, and outcomes. When $T \gg \tau$, this construction yields $T - \tau$ segments that partially overlap in time, providing additional training samples in this intrinsically data-scarce, single-chain setting.

However, these prefixes are *not* independent: successive segments share overlapping histories, so standard i.i.d. assumptions do not apply. In the next section, we introduce conditions under which these prefixes can be treated as *conditionally exchangeable* given an appropriate learned embedding. This enables regression-based estimation of CAPOs by pooling information across time without violating the dependence structure of the original process.

## 4 Identification and Estimation of CAPOs in Spatiotemporal Settings

Identification of CAPOs from observational data relies on standard causal inference assumptions. In our setting, these must be complemented by additional structure to handle the fact that we observe only a *single* spatiotemporal trajectory. Building on the prefix construction introduced above, we impose conditions that render these overlapping segments *conditionally exchangeable*, enabling principled pooling of information across time.

**Assumption 1** (Causal Inference Assumptions). *We assume: (Consistency)* $\mathbf{Y}_{t+\tau} = \mathbf{Y}_{t+\tau}[\mathbf{a}_{t:t+\tau-1}]$ *whenever the observed sequence of treatments* $\mathbf{A}_{t:t+\tau-1}$ *satisfies* $\mathbf{A}_{t:t+\tau-1} = \mathbf{a}_{t:t+\tau-1}$; *(Positivity)* $P(A_{s,t} = a_{s,t} \mid \mathbf{H}_{1:t} = \mathbf{h}_{1:t}) > 0$ *for any* $a_{s,t} \in \{0,1\}$ *and feasible realization of history* $\mathbf{h}_{1:t}$; *(Sequential Unconfoundedness)* $\mathbf{Y}_{t+1:T}[\mathbf{a}_{t+1:T}] \perp \mathbf{A}_t \mid \mathbf{H}_{1:t}, \forall \mathbf{a}_{t+1:T} \in \{0,1\}^{T-t}$, i.e. *at each time step* $t$, *the treatment assignment is independent of future potential outcomes.*

**Assumption 2** (Representation-Based Time Invariance). *There exists a function (or embedding)* $\phi : \mathcal{H} \times \mathcal{A} \rightarrow \mathcal{Z} \subseteq \mathbb{R}^h$ *that maps* $(\mathbf{H}_{1:t}, \mathbf{A}_t)$ *to a finite-dimensional representation such that once we condition on* $z = \phi(\mathbf{H}_{1:t}, \mathbf{A}_t)$, *the distribution* $(\mathbf{X}_{t+1}, \mathbf{Y}_{t+1})$ *does not explicitly depend on* $t$. *Formally, for any* $t, t' \in \{1, \ldots, T\}$ *and* $z \in \mathcal{Z}$, *we have:*

$$p(\mathbf{X}_{t+1}, \mathbf{Y}_{t+1} \mid \phi(\mathbf{H}_{1:t}, \mathbf{A}_t) = z) = p(\mathbf{X}_{t'+1}, \mathbf{Y}_{t'+1} \mid \phi(\mathbf{H}_{1:t'}, \mathbf{A}_{t'}) = z).$$

Assumption 1 is a standard set of requirements in longitudinal causal inference settings (e.g., [7, 18, 24, 28, 35, 36]). Assumption 2 is specific to the single-time series setting, where pooling information across time is essential to enable estimation. We note that the single time-series setting frequently arises in causal inference, where assumptions such as stationarity or strict time homogeneity enable consistent estimation [8, 31, 49]. In contrast, our representation-based time invariance is *weaker*: rather than requiring $\mathbf{X}_t, \mathbf{Y}_t$ themselves to have a time-invariant distribution, we only assume that, once the history is summarized by $\phi(\mathbf{H}_{1:t}, \mathbf{A}_t)$, the transition to $(\mathbf{X}_{t+1}, \mathbf{Y}_{t+1})$ follows a single shared mechanism. This approach aligns with modern time-series causal inference that learn time-invariant latent embeddings to pool information across time steps [18, 24, 26], thus leveraging more data for a single, stable representation rather than time-dependent parameters.

Under Assumption 2, conditioning on $\phi(\mathbf{H}_{1:t}, \mathbf{A}_t)$ removes explicit dependence on $t$, such that

$$\mathbb{E}_{\mathbf{P}}[\mathbf{Y}_{t+\tau} \mid \phi(\mathbf{H}_{1:t}, \mathbf{A}_t)]$$

represents a shared conditional expectation across all prefix segments. In this view, $t$ indexes the segment's position rather than a distinct distribution. Pooling over $t$ thus yields $T - \tau$ approximately exchangeable segments from a single trajectory, enabling regression-based estimation of future outcomes from embedded histories.

### 4.1 Identification via Representation-Based G-Computation

Given $\mathbf{P}_t^\tau$, we next show how to identify CAPOs from observational data. For horizons $\tau \geq 2$, *future* covariates and outcomes (*i.e.* $\mathbf{X}_{t+1:t+\tau-1}, \mathbf{Y}_{t+1:t+\tau-1}$) can influence subsequent treatments, inducing time-varying confounding [13]. Such feedback violates standard "condition-on-history" adjustments and leads to biased estimates. Figure 1 illustrates these dependencies by contrasting observational data (left) and hypothetical interventions (right) for $\tau = 2$. By contrast, when $\tau = 1$, conditioning on $\mathbf{H}_{1:t}$ is sufficient under standard assumptions, as no future confounders intervene between $\mathbf{A}_t$ and $\mathbf{Y}_{t+1}$. Formally, the following naive identification fails to hold for $\tau > 1$:

$$\mathbb{E}[\mathbf{Y}_{t+\tau}[\mathbf{a}_{t:t+\tau-1}] \mid \mathbf{H}_{1:t} = \mathbf{h}_{1:t}] \neq \mathbb{E}[\mathbf{Y}_{t+\tau} \mid \mathbf{H}_{1:t} = \mathbf{h}_{1:t}, \mathbf{A}_{t:t+\tau-1} = \mathbf{a}_{t:t+\tau-1}] \qquad (2)$$

To correct this bias, we adapt *regression-based iterative G-computation* [3, 35] to the spatiotemporal setting, yielding a principled adjustment procedure for evolving confounders and valid CAPO estimation. We formalize this connection in the following result:

**Theorem 1** (Identification with G-Computation). *Assume that Assumption 1 and Assumption 2 hold. Further, let $\mathbf{H}^{\mathbf{a}}_{1:t+k} := (\mathbf{X}_{1:t+k}, [\mathbf{A}_{1:t-1}, \mathbf{a}_{t:t+k-1}], \mathbf{Y}_{1:t+k})$ denote the history where observed treatments from time $t$ onward are replaced by $\mathbf{a}_{t:t+k-1}$. Define recursively:*

$$Q_\tau(\mathbf{H}_{1:t+\tau-1}, \mathbf{A}_{t+\tau-1}) = \mathbb{E}_{\mathbf{P}}[\mathbf{Y}_{t+\tau} \mid \phi(\mathbf{H}_{1:t+\tau-1}, \mathbf{A}_{t+\tau-1})]$$
$$Q_{\tau-1}(\mathbf{H}_{1:t+\tau-2}, \mathbf{A}_{t+\tau-2}) = \mathbb{E}_{\mathbf{P}}[Q_\tau(\mathbf{H}^{\mathbf{a}}_{1:t+\tau-1}, \mathbf{a}_{t+\tau-1}) \mid \phi(\mathbf{H}_{1:t+\tau-2}, \mathbf{A}_{t+\tau-2})]$$
$$\cdots$$
$$Q_1(\mathbf{H}_{1:t}, \mathbf{A}_t) = \mathbb{E}_{\mathbf{P}}[Q_2(\mathbf{H}^{\mathbf{a}}_{1:t+1}, \mathbf{a}_{t+1}) \mid \phi(\mathbf{H}_{1:t}, \mathbf{A}_t)]$$

*Then $\mathbb{E}[\mathbf{Y}_{t+\tau}[\mathbf{a}_{t:t+\tau-1}] \mid \mathbf{H}_{1:t} = \mathbf{h}_{1:t}] = Q_1(\mathbf{h}_{1:t}, \mathbf{a}_t)$.*

We provide a proof of Theorem 1 in Appendix B. This result naturally motivates a recursive regression approach for spatiotemporal CAPO estimation, fitting each $Q_k(\cdot)$ in reverse order and substituting interventional treatments where required.

### 4.2   Estimation via Iterative G-Computation

While Theorem 1 motivates a recursive regression algorithm for each $Q_k$ ($k = 1, \ldots, \tau$), only $Q_\tau$ can be directly estimated from the prefix data. At the next step, $Q_{\tau-1}$ depends on $Q_\tau\big(\mathbf{H}^{\mathbf{a}}_{1:t+\tau-1}, \mathbf{a}_{t+\tau-1}\big)$—where the observed treatments $\mathbf{A}_{t:t+\tau-1}$ are replaced by $\mathbf{a}_{t:t+\tau-1}$—but such substituted outcomes are not observed in the prefix data. Therefore, for $k < \tau$, we propose a procedure where we generate *pseudo-outcomes* by predicting with the previously learned $\widehat{Q}_{k+1}$. Going forward, we use $\widehat{F}$ to denote any quantity $F$ estimated from data. Formally, let $\phi \in \Phi$ be an embedding satisfying Assumption 2, and let $\mathcal{Q}$ be our function class for $Q_k$. We learn the sequence $\widehat{Q}_\tau, \ldots, \widehat{Q}_1$ from prefix data $\{\mathbf{P}^\tau_t : t = 1, \ldots, T - \tau\}$, via:

1. **Initialization.** Fit $\widehat{Q}_\tau$ to predict $\mathbf{Y}_{t+\tau}$ from the prefix embedding $\phi(\mathbf{H}_{1:t+\tau-1}, \mathbf{A}_{t+\tau-1})$.
2. **Backward recursion.** For $k = \tau - 1, \ldots, 1$:
   (a) *Substitute interventions.* For each prefix $\mathbf{P}^\tau_t$, replace $\mathbf{A}_{t+k}$ by the interventional $\mathbf{a}_{t+k}$ to form the modified history $\mathbf{H}^{\mathbf{a}}_{1:t+k}$.
   (b) *Generate pseudo-outcomes.* Let $\widetilde{Y}_{t+k+1} = \widehat{Q}_{k+1}\big(\mathbf{H}^{\mathbf{a}}_{1:t+k}, \mathbf{a}_{t+k}\big)$, where $\widehat{Q}_{k+1}$ was learned in the previous step. These $\widetilde{Y}_{t+k+1}$ act as surrogates for $\mathbf{Y}_{t+k+1}$ in the prefix data.
   (c) *Fit $\widehat{Q}_k$.* Regress $\widetilde{Y}_{t+k+1}$ on the current embedding $\phi\big(\mathbf{H}_{1:t+k-1}, \mathbf{A}_{t+k-1}\big)$ to learn $\widehat{Q}_k \in \mathcal{Q}$.
3. **Final step.** Given a new history $\mathbf{h}_{1:t}$ and an interventional path $\mathbf{a}_{t:t+\tau-1}$, we predict

$$\widehat{\mathbb{E}}_{\mathbf{P}}[\mathbf{Y}_{t+\tau}[\mathbf{a}_{t:t+\tau-1}] \mid \phi(\mathbf{H}_{1:t}, \mathbf{a}_t) = \phi(\mathbf{h}_{1:t}, \mathbf{a}_t)] = \widehat{Q}_1\big(\mathbf{h}_{1:t}, \mathbf{a}_t\big).$$

The iterative regression procedure yields consistent CAPO estimates provided each stage $Q_k$ is estimated consistently from data [22]. Informally, if the learned embedding $\widehat{\phi}$ converges to the true time-invariant representation $\phi$, and small perturbations in $\phi$ or $\widehat{Q}_k$ lead to proportionally small changes in predictions, then the overall recursive estimator remains consistent. These regularity conditions—formalized through uniform stochastic equicontinuity—are detailed in Appendix C. Formally, we state the following theorem:

**Theorem 2** (Consistency of Iterative G-Computation in Spatiotemporal Settings). *Assume Assumptions 1 and 2 and that (a) the learned embedding $\widehat{\phi}$ is $L_2$-consistent for $\phi$, and (b) each regression head $\widehat{Q}_k$ consistently estimates $Q_k$ and is uniformly well-behaved[1] on $\mathrm{Im}\,\phi$ (intuitively, small input perturbations induce small output changes). Let $\mathbf{Z}_k := (\mathbf{H}_{1:t+k}, \mathbf{A}_{t+k})$ denote the history–action pair at step $k$. Then*

$$\big\|\widehat{Q}_1(\mathbf{Z}_0; \widehat{\phi}) - Q_1(\mathbf{Z}_0; \phi)\big\|_2 = o_p(1),$$

*so the recursive estimator $\widehat{Q}_1$ of the CAPO is probabilistically consistent.*

We provide a proof of Theorem 2 in Appendix C. In the following section, we instantiate this procedure in our **GST-UNet** architecture, illustrating how to incorporate spatial dependencies and interference into $\phi$ and each $Q_k$, and implement a streamlined, end-to-end training strategy that unifies history embeddings and outcome predictions.

---

[1]We formalize "well-behaved" via uniform stochastic equicontinuity and continuity in Appendix C.

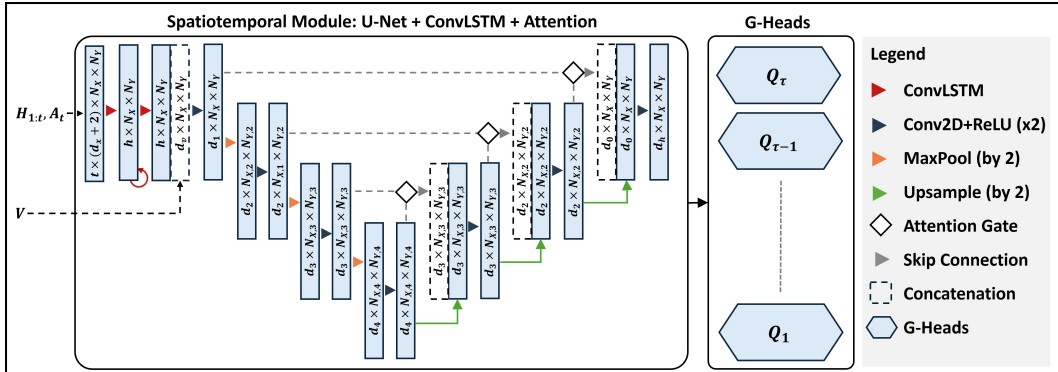

Figure 2: Overview of the GST-UNet architecture. The spatiotemporal learning module (left) is a U-Net augmented with a ConvLSTM layer and attention gates. Its final feature map is passed to a set of *G-heads* (right), where each G-head $Q_k$ implements iterative G-computation (see Algorithm 1).

## 5    GST-UNet Implementation

The theoretical results above establish how CAPOs can be identified and consistently estimated from a single spatiotemporal trajectory. We now provide a concrete neural implementation of this procedure. **GST-UNet** instantiates the iterative G-computation framework with a spatiotemporal deep architecture that embeds strong inductive biases—locality, translation invariance, and temporal smoothness—well suited to data-scarce settings. While alternative backbones could be employed, our U-Net with ConvLSTM and attention offers a natural choice for learning stable, history-invariant representations that satisfy Assumption 2. We now describe the architecture and the training procedure that realizes the GST-UNet (Algorithm 1).

### 5.1    Model Architecture

The **GST-UNet** consists of two main components:

1. **Spatiotemporal Learning Module:** a U-Net-based network augmented with ConvLSTM and attention gates for spatiotemporal processing.
2. **Neural Causal Module:** $\tau$ G-computation heads, each mapping the spatiotemporal features to the final outcome predictions in the iterative procedure.

We illustrate the GST-UNet architecture in Figure 2 and describe its main components below.

**Spatiotemporal Learning Module.** *(1) Spatial Module.* While our framework is agnostic to the choice of spatiotemporal learning module, we adopt a U-Net with ConvLSTM and attention due to its strong performance in data-scarce regimes. To efficiently process high-dimensional spatial data, we employ U-Net [37], a fully convolutional architecture originally developed for biomedical image segmentation. It employs an encoder-decoder design with skip connections: the encoder progressively downsamples the spatial grid through convolution and pooling, while the decoder upsamples it back to the original resolution, merging encoder features at each scale. *(2) Temporal Module.* U-Net has limitations in capturing temporal information. To address this, we integrate a Convolutional Long Short-Term Memory (ConvLSTM) layer [40] to the U-Net encoder. This module captures temporal dependencies by maintaining a hidden state across time steps while aggregating spatial information through convolutions. After computing the final ConvLSTM state, we append static (time-invariant) covariates **V** as additional feature channels, ensuring the subsequent U-Net encoder-decoder has direct access to both temporal dynamics and static location-specific information. In the decoder, we incorporate *attention gates* [29] to selectively highlight relevant spatial regions, refining skip connections and emphasizing critical global or local patterns. The embedding module ultimately produces a $d_h$-dimensional feature map of size $N_X \times N_Y$, capturing essential spatiotemporal context—including interference, spatial confounding, and static covariates—for downstream G-computation.

**Neural Causal Module.** We attach $\tau$ *G-computation heads* to the U-Net's final feature maps, corresponding to the $Q_k$ estimators in the iterative procedure (see Section 4.2). Each head can be a small convolutional module or a simple feed-forward network, depending on how much spatial structure

---

**Algorithm 1** GST-UNet Training and Inference

---

1: **Input:** Horizon $\tau$, prefixes $\{\mathbf{P}_t^\tau\}_{t=1}^{T-\tau}$, interventions $\mathbf{a}_{t:t+\tau-1}$, curriculum $\alpha_k^{(e)}$, total epochs $E$.
2: **Initialize:** parameters $\theta$ (U-Net embedding + G-heads).
3: **for** $e = 1 \ldots E$ **do**
4:      **for** $k = \tau \ldots 1$ **do**
5:          **(Supervision)** For each prefix $i$, predict outcomes $\widehat{Y}_{t+k}^{(i)} = Q_k\big(\phi(\mathbf{H}_{1:t+k-1}^{(i)}, \mathbf{A}_{t+k-1}^{(i)}); \theta\big)$.
6:          **(Generation (detached))** For each prefix $i$, generate pseudo-outcomes:

$$\widetilde{Y}_{t+k+1}^{(i)} = \begin{cases} Q_{k+1}\Big(\phi\big((\mathbf{H}_{1:t+k}^{\mathbf{a}})^{(i)}, \mathbf{a}_{t+k}^{(i)}); \theta\big), & k < \tau, \\ Y_{t+\tau}^{(i)}, & k = \tau, \end{cases}$$

         where the observed $\mathbf{A}_{t:t+k-1}$'s were replaced with $\mathbf{a}_{t:t+k-1}$ in $\mathbf{H}_{1:t+k}^{\mathbf{a}}$.
7:      **(Loss aggregation)** Compute the MSE loss $\mathcal{L}(\theta; e) = \frac{1}{\tau} \sum_{k=1}^{\tau} \alpha_k^{(e)} \sum_i \big(\widehat{Y}_{t+k}^{(i)} - \widetilde{Y}_{t+k+1}^{(i)}\big)^2$.
8:      **(Backward pass)** Update $\theta$ by backpropagation.
9: **(Inference)** Given a $\mathbf{h}_{1:t}$, return $Q_1(\phi(\mathbf{h}_{1:t}, \mathbf{a}_t); \widehat{\theta})$.

---

remains to be captured. The information flow at the G-computation heads proceeds as follows: each head $Q_k$ ($k = 1, \ldots, \tau$) receives the $d_h \times N_X \times N_Y$ U-Net embedding $\widehat{\phi}\big(\mathbf{H}_{1:t+k-1}, \mathbf{A}_{t+k-1}\big)$ (encompassing spatiotemporal and static context) and outputs an $N_X \times N_Y$ prediction for that time step. We refer to this as the *supervision step*, since $Q_\tau$ compares its predictions to the *real* observed outcomes $\mathbf{Y}_{t+\tau}$, anchoring the model in genuine data, while each $Q_{k<\tau}$ compares its predictions to pseudo-outcomes $\widetilde{\mathbf{Y}}_{t+k+1}$ provided by $\widehat{Q}_{k+1}$. These pseudo-outcomes arise in a subsequent *generation step*, wherein $Q_{k+1}$ processes the intervened history $\widehat{\phi}\big(\mathbf{H}_{1:t+k}^{\mathbf{a}}, \mathbf{a}_{t+k}\big)$ in a *detached* forward pass (so $\widehat{Q}_{k+1}$ is not updated by $Q_k$'s loss), thereby creating surrogate targets for $Q_k$. This procedure realizes the iterative G-computation logic from Section 4.2, enabling GST-UNet to estimate future outcomes under various counterfactual treatments. By separating the spatiotemporal embedding from the G-heads, we maintain a common representation for all prefix data (see Assumption 2) and flexibly capture interference and spatial confounding. Each G-head enforces the proper temporal adjustments to yield bias-free counterfactual inference.

## 5.2 Training and Inference

While each G-head $Q_k$ could be trained sequentially–from $Q_\tau$ down to $Q_1$–by passing pseudo-outcomes backward through time, this creates a conflict when all heads share the same U-Net embedding $\phi$. Specifically, each $Q_k$ may push $\phi$ toward optimizing its own objective, resulting in misaligned training signals and unstable learning.

**Joint Loss and Multi-Task Training.** To address this issue, we employ a *joint* (or *multi-task*) training approach [9, 15] by aggregating the loss terms from all G-heads into a single objective, then backpropagating once per batch. Concretely, for each head $Q_k$, let $\widetilde{Y}_{t+k+1}$ be the *real* outcomes if $k = \tau$ or *pseudo-outcomes* (generated by $\widehat{Q}_{k+1}$) if $k < \tau$. Our head-specific loss is a mean squared error (MSE) over all prefix samples:

$$\mathcal{L}_k(\theta) = \sum_{i=1}^{T-\tau} \Big[ Q_k\big(\phi(\mathbf{H}_{1:t+k-1}^{(i)}, \mathbf{A}_{t+k-1}^{(i)}); \theta\big) - \widetilde{Y}_{t+k+1}^{(i)} \Big]^2,$$

where $\theta$ encompasses *all* model parameters (the shared U-Net embedding $\phi$ and the G-heads $Q_k$).

Let $\alpha_k^{(e)}$ denote a *head-weight* for epoch $e$. We then form the overall training objective at epoch $e$ by

$$\mathcal{L}(\theta; e) = \frac{1}{\tau} \sum_{k=1}^{\tau} \alpha_k^{(e)} \mathcal{L}_k(\theta). \tag{3}$$

By summing the losses and performing a single backward pass, we learn a common embedding $\widehat{\phi}$ that balances the needs of all G-heads, rather than fitting each head separately.

**Curriculum Training.** A naive implementation of Eq. (3)–where each G-head is given equal weight– can be suboptimal: early in training, $Q_\tau$ (which sees real data) is inaccurate, and the pseudo-outcomes

generated for $Q_{k<\tau}$ are effectively noise. Consequently, $Q_1, \ldots, Q_{\tau-1}$ may overfit to poor targets before $Q_\tau$ has converged, leading to suboptimal solutions. To mitigate this, we employ a *curriculum* training approach [5], gradually increasing the loss weight of earlier heads as $Q_\tau$ improves.

While many curricula are possible, we adopt a simple scheme controlled by a single hyperparameter $e_c$ (the "curriculum period") so we can readily tune it. Let $p(e) = \min\{\tau, \lceil e/e_c \rceil\}$, which indexes a "phase" based on the current epoch $e$. We then define

$$\alpha_k^{(e)} = \begin{cases} 1/p(e), & \text{if } k \in \{\tau, \tau-1, \ldots, \tau-p(e)+1\}, \\ 0, & \text{otherwise.} \end{cases}$$

Hence, during epochs $1 \leq e \leq e_c$ (phase $p(e) = 1$), only $Q_\tau$ is active with $\alpha_\tau^{(e)} = 1$; in the next interval $e_c < e \leq 2e_c$ (phase $p(e) = 2$), $Q_\tau$ and $Q_{\tau-1}$ each have weight $1/2$, and so on until all heads are active with uniform weight $1/\tau$. For $e > \tau e_c$, training continues with $\alpha_k^{(e)} = 1/\tau$ for all heads. This schedule ensures $Q_\tau$ becomes reasonably accurate before earlier heads rely on its pseudo-outcomes. The hyperparameter $e_c$ controls the pacing, helping prevent early training noise.

We also adopt standard neural network practices, including mini-batch optimization and early stopping, to stabilize training and mitigate overfitting. At *inference* time, given a new history $\mathbf{h}_{1:t}$ and an interventional sequence $\mathbf{a}_{t:t+\tau-1}$, we compute $\widehat{Q}_1(\phi(\mathbf{h}_{1:t}, \mathbf{a}_t); \theta)$ as our target CAPO estimate. We sketch the overall training and inference procedure in Algorithm 1.

## 6 Experiments

We evaluate the proposed GST-UNet framework through two applications. First, we simulate synthetic data that incorporates key spatiotemporal causal inference challenges: interference, spatial confounding, temporal carryover, and time-varying confounding. Using this synthetic data generation process (DGP), we compare the GST-UNet algorithm against several baselines. Next, we demonstrate the utility of GST-UNet on a real-world dataset analyzing the impact of wildfire smoke on respiratory hospitalizations during the 2018 California Camp Fire.

Additional details–including exact simulation parameters, model architecture and execution setups, hyperparameter selection strategies, and validation procedures–can be found in Appendix D. Replication code is available at `https://github.com/moprescu/GSTUNet`.

### 6.1 Synthetic Data

We generate $T = 200$ time steps of a $64 \times 64$ ($N_X \times N_Y$) grid of observational data using the following data generating process (DGP):

$$\mathbf{X}_t = \alpha_0 + \alpha_1 \mathbf{X}_{t-1} + \alpha_2 \mathbf{A}_{t-1} + \alpha_3 (K_X * \mathbf{X}_{t-1}) + \epsilon_X,$$

$$\mathbf{A}_t \sim \text{Bern}\Big(\sigma\big(\beta_1\big(\beta_0 + \tfrac{1}{L}\sum_{l=0}^{L-1} K_A * \mathbf{X}_{t-l}\big)\big)\Big),$$

$$\mathbf{Y}_t = \gamma_0 + \gamma_1\big(K_{YA} * \mathbf{A}_{t-1}\big) + \gamma_2 \tfrac{1}{L}\sum_{l=1}^{L}\big(K_{YX} * \mathbf{X}_{t-l}\big) + \gamma_3 \mathbf{Y}_{t-1} + \epsilon_Y,$$

where $d_X = 1$, "$*$" denotes a $3 \times 3$ spatial convolution over the $N_X \times N_Y$ grid, and $\epsilon_X, \epsilon_Y \sim \mathcal{N}(0, 1)$ are i.i.d. noise. Each kernel $K_X, K_A, K_{YA}, K_{YX}$ encodes a local advection–diffusion process that mimics wind-driven pollutant transport, with interventions $\mathbf{A}_t$ injecting additional emissions that propagate through the same kernel. This physically realistic setup produces **interference**, **spatial confounding**, and **temporal carryover**—the three challenges GST-UNet is designed to address. Each equation is evaluated at every spatial location, so $\mathbf{X}_t$, $\mathbf{A}_t$, and $\mathbf{Y}_t$ are $N_X \times N_Y$ matrices. Here, $\mathbf{X}_t$ acts as a *time-varying confounder*: its past influences both $\mathbf{A}_t$ and $\mathbf{Y}_t$, while current interventions $\mathbf{A}_t$ affect future $\mathbf{X}_{t+1}$. For example, $\mathbf{A}_t$ may represent regulatory actions, $\mathbf{X}_t$ air quality, and $\mathbf{Y}_t$ health outcomes—capturing feedback from policy to exposure to outcome, and back to future policy.

We vary $\beta_1$ to control time-varying confounding: when $\beta_1 = 0$, $\mathbf{X}_t$ does not affect $\mathbf{A}_t$, eliminating confounding; larger values increase its strength. For each $\beta_1$, we generate 50 test trajectories from random initial states, fix their histories, and simulate 100 $\tau$-step counterfactual futures to estimate true

Table 1: RMSE $\pm$ SD across test trajectories. Bold indicates lowest error per column; color shows improvement (RMSE **decrease** or **increase**) over best baseline (excluding ablations).

| $\tau$ | Model | $\beta_1 = 0.0$ | $\beta_1 = 0.5$ | $\beta_1 = 1.0$ | $\beta_1 = 1.5$ | $\beta_1 = 2.0$ |
|---|---|---|---|---|---|---|
| 5 | UNet+ | **0.28 ± 0.00** | 0.36 ± 0.00 | 0.54 ± 0.01 | 0.71 ± 0.01 | 0.81 ± 0.01 |
| | STCINet | 0.29 ± 0.00 | 0.38 ± 0.01 | 0.62 ± 0.01 | 0.80 ± 0.01 | 0.90 ± 0.01 |
| | IPWUNet | 0.60 ± 0.01 | 0.58 ± 0.01 | 0.58 ± 0.01 | 0.59 ± 0.01 | 0.59 ± 0.01 |
| | GST-UNet w/o Attention | 0.50 ± 0.00 | 0.46 ± 0.00 | 0.51 ± 0.00 | 0.45 ± 0.01 | 0.47 ± 0.01 |
| | GST-UNet w/o Curriculum | 0.69 ± 0.00 | 0.64 ± 0.00 | 0.63 ± 0.00 | 0.61 ± 0.01 | 0.61 ± 0.01 |
| | **GST-UNet** | 0.33 ± 0.00 | **0.35 ± 0.00** | **0.40 ± 0.00** | **0.44 ± 0.00** | **0.40 ± 0.01** |
| | | (+17.9%) | (-2.7%) | (-21.6%) | (-25.4%) | (-32.2%) |
| 10 | UNet+ | **0.28 ± 0.00** | 0.61 ± 0.00 | 1.18 ± 0.00 | 1.45 ± 0.00 | 1.71 ± 0.01 |
| | STCINet | 0.31 ± 0.00 | 0.68 ± 0.00 | 1.25 ± 0.00 | 1.47 ± 0.01 | 1.60 ± 0.01 |
| | IPWUNet | 0.78 ± 0.01 | 0.80 ± 0.01 | 0.96 ± 0.01 | 1.19 ± 0.02 | 1.08 ± 0.01 |
| | GST-UNet w/o Attention | 0.42 ± 0.00 | 0.60 ± 0.00 | 0.61 ± 0.00 | 0.79 ± 0.01 | 1.07 ± 0.01 |
| | GST-UNet w/o Curriculum | 0.62 ± 0.00 | 0.88 ± 0.00 | 1.02 ± 0.00 | 1.08 ± 0.01 | 1.12 ± 0.01 |
| | **GST-UNet** | 0.38 ± 0.00 | **0.55 ± 0.00** | **0.68 ± 0.00** | **0.73 ± 0.01** | **0.85 ± 0.01** |
| | | (+35.7%) | (-9.8%) | (-29.2%) | (-38.7%) | (-21.3%) |

CAPOs, with $\tau \in \{5, 10\}$. We compare GST-UNet against three baselines: (i) **UNet+**, which uses a U-Net + ConvLSTM + Attention backbone with $A_t$ as an input channel but performs no iterative adjustment; (ii) **STCINet** [1], which estimates direct and indirect effects without modeling time-varying confounding; and (iii) **IPWUNet**, an inverse-propensity-weighting variant that reweights pseudo-outcomes using a UNet-style propensity estimator but cannot correct for spatial interference (details in Appendix D). We also test ablations of GST-UNet without curriculum or attention. Table 1 shows that when $\beta_1 = 0$, UNet+ performs best—G-computation is unnecessary and adds noise. As $\beta_1$ increases, UNet+ and STCINet degrade sharply, while GST-UNet remains stable. IPWUNet shows some benefit but is biased even at $\beta_1 = 0$ due to uncorrected interference. GST-UNet consistently outperforms all baselines, demonstrating the value of iterative G-computation. Curriculum training substantially improves performance across horizons, while attention yields modest gains—consistent with our predominantly local dynamics. Additional ablation analyses, including neighbor aggregation experiments, are reported in Appendix D.

## 6.2 Impact of Wildfires on Respiratory Health

Wildfire smoke has been linked to short-term respiratory harms [10, 12, 23, 33, 34], with older adults especially vulnerable [14]. At the time this work was conducted (January 2025), a series of 14 destructive wildfires affected the Los Angeles metropolitan area and San Diego County in California, underscoring the urgency of understanding the health impacts of such events. In this study, we focus on a previous large-scale episode: the 2018 California wildfire season [45], which included the *Carr Fire* (July–August) and the *Camp Fire* (November) and significantly worsened air quality.

We use daily, county-level data from Letellier et al. [23] (see Appendix D.2), including PM$_{2.5}$, respiratory/cardiovascular hospitalizations, and weather variables (temperature, precipitation, humidity, radiation, wind), along with population estimates from the California Department of Finance. Each of the weather variables can be a *time-varying confounder*: weather conditions affect future smoke levels and health outcomes, while also being influenced by prior smoke levels.

We focus on weeks 20–48 (May 18–December 2, 2018), covering the Carr and Camp fires. Following standard practice, we label a county as "treated" on days with mean PM$_{2.5} > 10$ $\mu g/m^3$ and use raw hospitalization counts (rather than per-10,000 incidence, which can be unstable for small counties). We interpolate daily county-level data (treatment, outcome, five covariates) onto a $40 \times 44$ latitude–longitude grid, discarding cells outside California, yielding a spatiotemporal tensor of size $203 \times 7 \times 40 \times 44$. Interpolation ensures each grid cell approximates the region it overlaps (area-weighted), enabling the model to capture spatial gradients in PM$_{2.5}$, weather, and hospitalizations. We train GST-UNet with horizon $\tau = 10$, using the Carr Fire period (June–July) for validation, and generate counterfactual predictions for the Camp Fire peak, November 8–17. See Appendix D.2 for preprocessing and masking details.

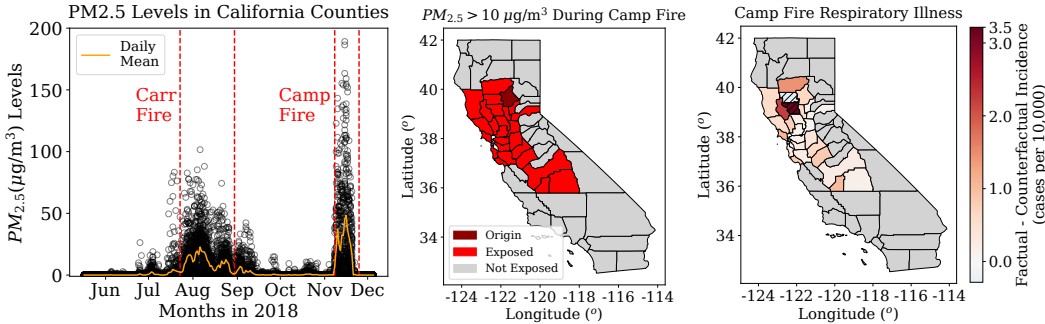

Figure 3: **(Left)** Daily $PM_{2.5}$ levels across California from May to December 2018, with red lines marking major wildfires. **(Center)** Counties exposed to average $PM_{2.5} > 10\,\mu g/m^3$ during the Camp Fire (red), origin county in dark red. **(Right)** Factual minus CAPO-predicted daily respiratory admissions during peak Camp Fire. Hashed areas indicate small-population counties ($< 30{,}000$).

Figure 3 (left) shows the rise in PM2.5 during the mid-late 2018 wildfire season; (center) highlights counties with daily PM2.5 $> 10, \mu g/m^3$. Using GST-UNet, we estimate daily CAPOs had the Camp Fire not occurred (i.e., setting $PM_{2.5} \leq 10, \mu g/m^3$ statewide). Figure 3 (right) compares these to factual daily incidence (hospitalizations per 10,000 residents). To reduce small-sample variability, we exclude counties with population below 30,000 (vs. $>70{,}000$ for others), marking them with hatching (see Appendix D.2). Over November 8–17, GST-UNet predicts **approximately 4,650 excess respiratory hospitalizations** (465/day) attributable to the Camp Fire, with the highest incidence near the fire source. This aligns with a 95% bootstrap confidence interval of $[1888, 6535]$. UNet+ yields a lower mean and higher uncertainty (3,981; $[-899, 5202]$), STCINet produces highly variable near-zero estimates (88; $[-3077, 3281]$), and IPWUNet gives implausibly high, near-constant values ($\sim$20,500), reflecting limitations of weighting under rare-event support. These results underscore GST-UNet's improved stability and accuracy in counterfactual estimation. Our findings are qualitatively consistent with Letellier et al. [23], who report 259 excess daily cases averaged over a longer, lower-intensity window (Nov 8-Dec 5). Overall, the GST-UNet captures spatiotemporal variation in smoke exposure and health outcomes, illustrating its promise for real-world causal inference in domains such as environmental health and policy.

## 7    Conclusion

We presented **GST-UNet**, a neural framework for spatiotemporal causal inference that combines U-Net–based representation learning with iterative G-computation to adjust for time-varying confounders. GST-UNet addresses key challenges such as interference, spatial confounding, temporal carryover, and time-varying feedback. We establish theoretical identification and consistency guarantees, validate performance in synthetic settings with controlled confounding, and demonstrate practical utility in estimating the impact of wildfire smoke exposure during the 2018 Camp Fire. Together, these results position GST-UNet as a **ready-to-use tool for practitioners**, offering reliable, interpretable causal estimates in complex spatiotemporal environments. We discuss limitations and broader impacts in Appendix E.

### Acknowledgements

We thank the anonymous reviewers for their thoughtful feedback and the constructive dialogue during the review process, which greatly strengthened the final version of this work. Miruna Oprescu and BNL team (D. Park, X. Luo, S. Yoo) were supported by the U.S. Department of Energy, Office of Science, Office of Advanced Scientific Computing Research, under Awards DE-SC0023112 and DE-SC0012704, respectively. Nathan Kallus was supported by the U.S. National Science Foundation under Grant No. 1846210. Part of this work was conducted while Miruna Oprescu was a research intern at Brookhaven National Laboratory. Any opinions, findings, and conclusions or recommendations expressed in this material are those of the author(s) and do not necessarily reflect the views of the U.S. Department of Energy or the U.S. National Science Foundation.

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

Table 2: Key differences between prior neural G-computation methods and GST-UNet.

| Aspect | Prior Neural G-Computation | GST-UNet (ours) |
|---|---|---|
| Data structure | Many independent temporal trajectories (e.g., patient sequences); no inter-unit interactions such as spatial dependence. | Single spatiotemporal chain where outcomes, covariates, and treatments evolve jointly across a lattice; strong spatial coupling and interference. |
| Encoder | RNN/Transformer over time only. | ConvLSTM-UNet encoder aggregates neighbour covariates/treatments before G-heads, capturing interference and spatial confounding. |
| Training | Standard end-to-end due to i.i.d trajectories; stability arises from large data rather than curriculum or spatial priors. | Curriculum-stabilized multi-head training for accurate pseudo-outcome generation under limited samples. |
| Theory | Classical G-formula under i.i.d. trajectories; no single-chain guarantees. | Identification (Theorem 1) and consistency (Theorem 3) under representation-based time-invariance for a single chain. |

# A  Extended Literature Review

**Classical Spatiotemporal Causal Inference.** Early spatiotemporal causal inference methods–including spatial econometrics [2], difference-in-differences [20], and synthetic controls [4]–provide useful frameworks for estimating treatment effects across regions but rely on strong assumptions such as parallel trends or stable treatment assignment. These approaches struggle with interference, nonlinear dependencies, and time-varying confounders, limiting their applicability in complex settings. More recent approaches for spatiotemporal causal inference handle time-varying confounding through inverse propensity weighting (IPW), typically by extending marginal structural models to the spatial or spatiotemporal domain. For instance, Papadogeorgou et al. [31] and Zhou et al. [49] employ IPW-style adjustments to estimate regional average treatment effects across space and time. However, these approaches cannot accommodate interference unless strong assumptions are made–e.g., defining a user-specified exposure mapping or restricting attention to hyper–local interactions (see also [11, 31, 44, 48]). Such simplifications may be ill-suited for real-world systems with rich spatial dependencies. Moreover, even recent advances in this space remain limited; as noted by Zhou et al. [49], the literature on spatiotemporal causal inference remains sparse, especially in settings with feedback loops or time-varying confounding.

**Machine Learning for Spatiotemporal Modeling.** Spatiotemporal predictive modeling has seen rapid progress with the rise of deep learning. Convolutional and recurrent neural networks are widely used for forecasting spatially indexed time series (e.g., weather or traffic) [40, 47], while graph-based methods (e.g., Graph WaveNet [46], Diffusion Convolutional RNN [25]) capture non-Euclidean spatial dependencies. Vision transformer variants, including Video Swin Transformers [27] and TimeSformer [6], extend attention-based models to spatiotemporal video data. These architectures can learn complex non-local interactions over space and time. However, such models are typically optimized for prediction tasks and do not include causal adjustments. Without mechanisms like propensity modeling or G-computation, they remain ill-equipped to estimate counterfactual outcomes or adjust for time-varying confounding. Some recent work integrates spatial representations for causal inference–e.g., Tec et al. [42] incorporate non-local confounders using a UNet-based model–but these methods do not explicitly model dependencies over time or adjust for time-varying confounders.

**Time-Series Causal Inference.** In the longitudinal domain, time-series causal inference has developed tools for handling temporal confounding using models such as marginal structural models [36], IPW-style estimation [26], and iterative G-computation [35]. Recent ML-based extensions include recurrent networks [7, 24, 39], Transformers [18, 28] and meta-learners [16]. However, all these methods assume access to independent time series–e.g., across units or patients–which allows for pooling across trajectories. These methods do not consider spatial dependencies, interference, or scenarios with a single observed spatiotemporal realization. As such, while they may handle time-varying confounding, they do not generalize to our setting. Table 2 summarizes the key methodological differences between GST-UNet and prior neural G-computation frameworks.

**Neural-Based Spatiotemporal Causal Inference.** There has been limited work on neural models that explicitly address spatiotemporal causal inference. Tec et al. [42] use a U-Net backbone to learn spatial representations for causal inference in air pollution studies but do not address time-varying confounding or feedback loops. Ali et al. [1] present a U-Net–based architecture for predicting direct and indirect effects in climate contexts, but primarily focus on forecasting rather than causal identification. While these works highlight growing interest in neural approaches to causal inference in spatiotemporal domains, none incorporate an iterative adjustment procedure like G-computation that handles time-varying confounders, leaving identification in these settings largely unaddressed.

**Our Contribution.** GST-UNet bridges these gaps by combining flexible spatiotemporal neural architectures with a theoretically grounded iterative G-computation framework. This allows valid estimation of potential outcomes in the presence of interference, spatial confounding, and time-varying confounding–without requiring practitioners to specify structural models or exposure mappings. To our knowledge, this is the first end-to-end framework to implement G-computation for causal inference over a single spatiotemporal trajectory. We integrate spatiotemporal processing via U-Nets and ConvLSTMs with a principled multi-head neural causal module, and we design a curriculum-based training strategy to stabilize learning of recursive pseudo-outcomes. Together, these components yield a ready-to-use tool for practitioners, with consistent identification guarantees and robust empirical performance. By abstracting away the modeling choices typically required in structural spatiotemporal methods, GST-UNet makes spatiotemporal causal estimation more accessible, interpretable, and reliable for real-world applications.

## B Proof of Theorem 1

We aim to show that under Assumption 1 and Assumption 2, the CAPOs in Equation (1) can be identified recursively from a single time series via a sequence of conditional expectations.

**Step 1: Recursive decomposition for the intractable expectation** We first demonstrate the recursive decomposition of the intractable expectation in the CAPO definition (Equation (1)). While this expectation is theoretically well-defined, it cannot be directly estimated in practice due to the limited availability of data. Specifically, we only observe a single time series, meaning we have just one sample of the history at time $t + \tau$ for each $t$. Nevertheless, as we will show, we can convert these expectations into expectations over prefix-based segments that allow us to estimate these quantities from the data.

Starting from $\mathbb{E}[\mathbf{Y}_{t+\tau}[\mathbf{a}_{t:t+\tau-1}] \mid \mathbf{H}_{1:t} = \mathbf{h}_{1:t}]$, we have:

$$\mathbb{E}[\mathbf{Y}_{t+\tau}[\mathbf{a}_{t:t+\tau-1}] \mid \mathbf{H}_{1:t} = \mathbf{h}_{1:t}]$$
$$= \mathbb{E}[\mathbf{Y}_{t+\tau}[\mathbf{a}_{t:t+\tau-1}] \mid \mathbf{H}_{1:t} = \mathbf{h}_{1:t}, \mathbf{A}_t = \mathbf{a}_t]$$
$$\text{(Sequential ignorability and positivity (Assumption 1))}$$
$$= \mathbb{E}\big[\mathbb{E}[\mathbf{Y}_{t+\tau}[\mathbf{a}_{t:t+\tau-1}] \mid \mathbf{H}^{\mathbf{a}}_{1:t+1}] \mid \mathbf{H}_{1:t} = \mathbf{h}_{1:t}, \mathbf{A}_t = \mathbf{a}_t\big] \qquad \text{(Law of total probability)}$$
$$= \mathbb{E}\big[\mathbb{E}[\mathbf{Y}_{t+\tau}[\mathbf{a}_{t:t+\tau-1}] \mid \mathbf{H}^{\mathbf{a}}_{1:t+1}, \mathbf{A}_{t+1} = \mathbf{a}_{t+1}] \mid \mathbf{H}_{1:t} = \mathbf{h}_{1:t}, \mathbf{A}_t = \mathbf{a}_t\big]$$
$$\text{(Sequential ignorability and positivity)}$$
$$= \mathbb{E}\Big[\mathbb{E}\big[\mathbb{E}[\mathbf{Y}_{t+\tau}[\mathbf{a}_{t:t+\tau-1}] \mid \mathbf{H}^{\mathbf{a}}_{1:t+2}] \,\big|\, \mathbf{H}^{\mathbf{a}}_{1:t+1}, \mathbf{A}_{t+1} = \mathbf{a}_{t+1}\big] \,\Big|\, \mathbf{H}_{1:t} = \mathbf{h}_{1:t}, \mathbf{A}_t = \mathbf{a}_t\Big]$$
$$\text{(Law of total probability)}$$
$$= \mathbb{E}\Big[\mathbb{E}\big[\mathbb{E}[\mathbf{Y}_{t+\tau}[\mathbf{a}_{t:t+\tau-1}] \mid \mathbf{H}^{\mathbf{a}}_{1:t+2}, \mathbf{A}_{t+2} = \mathbf{a}_{t+2}] \,\big|\, \mathbf{H}^{\mathbf{a}}_{1:t+1}, \mathbf{A}_{t+1} = \mathbf{a}_{t+1}\big] \,\Big|\, \mathbf{H}_{1:t} = \mathbf{h}_{1:t}, \mathbf{A}_t = \mathbf{a}_t\Big]$$
$$\text{(Sequential ignorability and positivity)}$$

$$\dots$$

$$= \mathbb{E}\Big[\dots \mathbb{E}\big[\mathbb{E}[\mathbf{Y}_{t+\tau}[\mathbf{a}_{t:t+\tau-1}] \mid \mathbf{H}^{\mathbf{a}}_{1:t+\tau-1}, \mathbf{A}_{t+\tau-1} = \mathbf{a}_{t+\tau-1}]$$
$$\Big| \mathbf{H}^{\mathbf{a}}_{1:t+\tau-2}, \mathbf{A}_{t+\tau-2} = \mathbf{a}_{t+\tau-2}\big]$$
$$\Big| \dots$$
$$\Big| \mathbf{H}_{1:t} = \mathbf{h}_{1:t}, \mathbf{A}_t = \mathbf{a}_t\Big] \quad \text{(Sequential ignorability and positivity)}$$
$$= \mathbb{E}\Big[\dots \mathbb{E}\big[\mathbb{E}[\mathbf{Y}_{t+\tau} \mid \mathbf{H}^{\mathbf{a}}_{1:t+\tau-1}, \mathbf{A}_{t+\tau-1} = \mathbf{a}_{t+\tau-1}]$$

$$\big| \, \mathbf{H}^{\mathbf{a}}_{1:t+\tau-2}, \mathbf{A}_{t+\tau-2} = \mathbf{a}_{t+\tau-2} \big]$$

$$\big| \, \dots$$

$$\big| \, \mathbf{H}_{1:t} = \mathbf{h}_{1:t}, \mathbf{A}_t = \mathbf{a}_t \big] \qquad \text{(Consistency)}$$

Thus, if we had multiple spatiotemporal time-series samples, we could directly estimate this nested expression from data, since the right-hand side depends solely on observed quantities, ensuring identifiability.

**Step 2: From intractable to prefix-based expectations**  We now show how to estimate the nested expectations using the prefix data. First, by Assumption 2, we can rewrite the inner-most expectation as

$$\mathbb{E}[\mathbf{Y}_{t+\tau} \mid \mathbf{H}^{\mathbf{a}}_{1:t+\tau-1}, \mathbf{A}_{t+\tau-1} = \mathbf{a}_{t+\tau-1}] = \mathbb{E}_{\mathbf{P}}[\mathbf{Y}_{t+\tau} \mid \phi(\mathbf{H}^{\mathbf{a}}_{1:t+\tau-1}, \mathbf{a}_{t+\tau-1})]$$
$$= Q_\tau(\mathbf{H}^{\mathbf{a}}_{1:t+\tau-1}, \mathbf{a}_{t+\tau-1}). \qquad \text{(Definition of } Q_\tau)$$

Thus, by using Assumption 1, we can write this expectation over the prefix data which we have many samples of. Now consider the next nested expectation:

$$\mathbb{E}[Q_\tau(\mathbf{H}^{\mathbf{a}}_{1:t+\tau-1}, \mathbf{a}_{t+\tau-1}) \mid \mathbf{H}^{\mathbf{a}}_{1:t+\tau-2} = \mathbf{h}^{\mathbf{a}}_{1:t+\tau-2}, \mathbf{A}_{t+\tau-2} = \mathbf{a}_{t+\tau-2}]$$

$$= \int Q_\tau(\mathbf{h}^{\mathbf{a}}_{1:t+\tau-1}, \mathbf{a}_{t+\tau-1}) p(x_{t+\tau-1}, y_{t+\tau-1} \mid \mathbf{h}^{\mathbf{a}}_{t+\tau-2}, \mathbf{a}_{t+\tau-2}) d(x_{t+\tau-1}, y_{t+\tau-1})$$

$$= \int_P Q_\tau(\mathbf{h}^{\mathbf{a}}_{1:t+\tau-1}, \mathbf{a}_{t+\tau-1}) p(x_{t+\tau-1}, y_{t+\tau-1} \mid \phi(\mathbf{h}^{\mathbf{a}}_{t+\tau-2}, \mathbf{a}_{t+\tau-2})) d(x_{t+\tau-1}, y_{t+\tau-1})$$
$$\text{(Assumption 2)}$$

$$= \mathbb{E}_P[Q_\tau(\mathbf{H}^{\mathbf{a}}_{1:t+\tau-1}, \mathbf{a}_{t+\tau-1}) \mid \phi(\mathbf{H}^{\mathbf{a}}_{1:t+\tau-2}, \mathbf{A}_{t+\tau-2}) = \phi(\mathbf{h}^{\mathbf{a}}_{1:t+\tau-2}, \mathbf{a}_{t+\tau-2})]$$
$$= Q_{\tau-1}(\mathbf{h}^{\mathbf{a}}_{1:t+\tau-2}, \mathbf{a}_{t+\tau-2})$$

Tracing this argument recursively through the nested expectation in Step 1, we obtain:

$$\mathbb{E}[\mathbf{Y}_{t+\tau}[\mathbf{a}_{t:t+\tau-1}] \mid \mathbf{H}_{1:t} = \mathbf{h}_{1:t}] = Q_1(\mathbf{h}_{1:t}, \mathbf{a}_t),$$

as desired. Thus, $Q_1$ – which can be estimated from the prefix data – recovers the CAPOs, under our assumptions, even from a single chain.

## C   Consistency of the Iterative G-Computation Estimator

In this section, we state the conditions under which the iterative G-computation procedure in Section 4.2 yields a consistent estimator, and show that our implementation of the $Q_k$ estimators satisfies these conditions.

**Notation:** We denote the $L_2$ norm of a function $f$ as $\|f\|_2 := \mathbb{E}_P[f(X)^2]^{1/2}$, where the expectation is over the probability distribution $P$. The notation $\widehat{f}_n$ represents the estimated value of a parameter or function learned on $n$ data points, where $f$ is the true value. For a sequence of random variables $\{Z_n\}_{n\geq 1}$ we write $Z_n = o_p(1)$ if $\Pr(|Z_n| > \varepsilon) \to 0$ for every $\varepsilon > 0$, *i.e.* $Z_n \xrightarrow{p} 0$.

To begin, we introduce the following stochastic equicontinuity condition from [43]:

**Definition 1** (Stochastic equicontinuity [43, Def. 1.5.7])**.** *Let $(\mathcal{Z}, d)$ be a semi-metric space and $\{\widehat{f}_n\}_{n\geq 1} \subset \ell^\infty(\mathcal{Z})$ a sequence of random functions. It is* uniformly stochastically equi-continuous *if, for every $\epsilon > 0, \eta > 0$, there exists a $\delta > 0$ such that*

$$\limsup_{n\to\infty} P\Big( \sup_{d(z,z')\leq\delta} \big|\widehat{f}_n(z) - \widehat{f}_n(z')\big| > \epsilon \Big) < \eta.$$

Stochastic equicontinuity ensures that, with high probability, each estimator changes only slightly when its input is perturbed by a small amount. It is strictly weaker than global Lipschitz continuity – any family that is Lipschitz on a bounded domain with constants bounded in probability automatically satisfies Definition 1. We impose this condition in Theorem 3 so that the $o_p(1)$ error in the learned embedding propagates to only $o_p(1)$ errors in the G-heads, making the recursive estimator consistent.

The following theorem restates Theorem 2 from the main text in full detail and provides its proof.

**Theorem 3** (Consistency under Uniform Stochastic Equicontinuity). *Suppose the conditions of Theorem 1 hold, and let $\widehat{\phi}$ be a learned embedding. Define $\mathbf{Z}_k := (\boldsymbol{H}_{1:t+k}, \boldsymbol{A}_{t+k})$, and recursively define the learned estimators $\widehat{Q}_k(\mathbf{Z}_{k-1}; \widehat{\phi}) := \widehat{\mathbb{E}}_{\mathbf{P}}[\widehat{Q}_{k+1}(\mathbf{Z}_k; \widehat{\phi}) \mid \widehat{\phi}(\mathbf{Z}_k)]$ for $k = 1, \ldots, \tau$, with terminal condition $\widehat{Q}_{\tau+1}(\mathbf{Z}_\tau; \widehat{\phi}) = Y^{t+\tau}$. Assume that $\{\widehat{Q}_k\}_{k=1}^\tau$ are obtained via the iterative G-computation algorithm. If:*

*(i) $\|\widehat{\phi} - \phi\|_2 = o_p(1)$;*
*(ii) $\|\widehat{Q}_k(\mathbf{Z}_{k-1}; \phi) - Q_k(\mathbf{Z}_{k-1}; \phi))\|_2 = o_p(1)$ for all $k$;*
*(iii) for every $k$ the random maps $z \mapsto \widehat{Q}_k(h, a; z)$ are stochastically equicontinuous on $\operatorname{Im}\phi$ (Definition 1), and $Q_k(\cdot)$ is uniformly continuous there,*

*then*

$$\left\|\widehat{Q}_1(\mathbf{Z}_0; \widehat{\phi}) - Q_1(\mathbf{Z}_0; \phi)\right\|_2 = o_p(1).$$

*Thus the recursive G-computation estimator is (probabilistically) consistent.*

*Proof.* We proceed by reverse induction on $k$, starting from $k = \tau$ and working backward to $k = 1$. For each $k$, we aim to show:

$$\|\widehat{Q}_k(\mathbf{Z}_{k-1}; \widehat{\phi}) - Q_k(\mathbf{Z}_{k-1}; \phi)\|_2 = o_p(1).$$

**Base case** ($k = \tau$). By definition, $\widehat{Q}_{\tau+1}(\mathbf{Z}_\tau; \widehat{\phi}) = Y^{t+\tau}$, which is observed. Thus,

$$\widehat{Q}_\tau(\mathbf{Z}_{\tau-1}; \widehat{\phi}) = \widehat{\mathbb{E}}_{\mathbf{P}}[Y^{t+\tau} \mid \widehat{\phi}(\mathbf{Z}_\tau)] \quad \text{and} \quad Q_\tau(\mathbf{Z}_{\tau-1}; \phi) = \mathbb{E}[Y^{t+\tau} \mid \phi(\mathbf{Z}_\tau)].$$

We decompose the difference:

$$\left\|\widehat{Q}_\tau(\mathbf{Z}_{\tau-1}; \widehat{\phi}) - Q_\tau(\mathbf{Z}_{\tau-1}; \phi)\right\|_2 \le \underbrace{\left\|\widehat{Q}_\tau(\mathbf{Z}_{\tau-1}; \widehat{\phi}) - \widehat{Q}_\tau(\mathbf{Z}_{\tau-1}; \phi)\right\|_2}_{\Lambda_1} + \underbrace{\left\|\widehat{Q}_\tau(\mathbf{Z}_{\tau-1}; \phi) - Q_\tau(\mathbf{Z}_{\tau-1}; \phi)\right\|_2}_{\Lambda_2}.$$

Term $\Lambda_2$ is $o_p(1)$ by assumption (ii). Term $\Lambda_1$ converges to zero in probability due to (i) $\|\widehat{\phi} - \phi\|_2 = o_p(1)$ and (iii) stochastic equicontinuity of $\widehat{Q}_\tau$. Therefore,

$$\left\|\widehat{Q}_\tau(\mathbf{Z}_{\tau-1}; \widehat{\phi}) - Q_\tau(\mathbf{Z}_{\tau-1}; \phi)\right\|_2 = o_p(1).$$

**Inductive step.** Suppose for some $k + 1 \le \tau$ that

$$\left\|\widehat{Q}_{k+1}(\mathbf{Z}_k; \widehat{\phi}) - Q_{k+1}(\mathbf{Z}_k; \phi)\right\|_2 = o_p(1).$$

We now consider

$$\widehat{Q}_k(\mathbf{Z}_{k-1}; \widehat{\phi}) = \widehat{\mathbb{E}}_{\mathbf{P}}[\widehat{Q}_{k+1}(\mathbf{Z}_k; \widehat{\phi}) \mid \widehat{\phi}(\mathbf{Z}_k)], \quad Q_k(\mathbf{Z}_{k-1}; \phi) = \mathbb{E}[Q_{k+1}(\mathbf{Z}_k; \phi) \mid \phi(\mathbf{Z}_k)].$$

Again, decompose:

$$\left\|\widehat{Q}_k(\mathbf{Z}_{k-1}; \widehat{\phi}) - Q_k(\mathbf{Z}_{k-1}; \phi)\right\|_2 \le \left\|\widehat{Q}_k(\mathbf{Z}_{k-1}; \widehat{\phi}) - \widehat{Q}_k(\mathbf{Z}_{k-1}; \phi)\right\|_2$$
$$+ \left\|\widehat{Q}_k(\mathbf{Z}_{k-1}; \phi) - Q_k(\mathbf{Z}_{k-1}; \phi)\right\|_2.$$

The second term is $o_p(1)$ by assumption (ii). The first term is also $o_p(1)$ because $\widehat{\phi} \to \phi$ in $L_2$ and the stochastic equicontinuity of $\widehat{Q}_k$ ensures that perturbations in $\phi$ yield small changes in predictions uniformly over $\operatorname{Im}\phi$. Thus,

$$\|\widehat{Q}_k(\mathbf{Z}_{k-1}; \widehat{\phi}) - Q_k(\mathbf{Z}_{k-1}; \phi)\|_2 = o_p(1).$$

By induction, the result holds for all $k = \tau, \tau - 1, \ldots, 1$, and in particular:

$$\|\widehat{Q}_1(\mathbf{Z}_0; \widehat{\phi}) - Q_1(\mathbf{Z}_0; \phi)\|_2 = o_p(1).$$

Thus, the proof is now complete. $\square$

**Example 1** (Feed-forward or convolutional heads). *Suppose each G-computation head $Q_k(\cdot; z)$ is implemented as a depth-$d$ neural network*

$$\Psi(z) = W_d \sigma_{d-1}\big(\cdots \sigma_1(W_1 z)\big),$$

*where the activations $\sigma_\ell$ are Lipschitz continuous (e.g., ReLU, Leaky ReLU, SoftPlus, Tanh, Sigmoid, or ArcTan). If each layer weight satisfies a spectral norm bound $\|W_\ell\|_2 \leq \rho_\ell < \infty$, then $\Psi$ is globally Lipschitz on $\mathbb{R}^h$ with constant $L = \prod_\ell \rho_\ell$, and thus uniformly continuous on any compact subset. This implies the stochastic equicontinuity condition in Definition 1.*

*In practice, norm control can be enforced via weight decay, spectral normalization, or weight clipping during training. Similarly, the encoder output $\widehat{\phi}(H, A)$ can be bounded—e.g., through normalization or clipping—so its image lies in a compact subset of $\mathbb{R}^h$. Together, these ensure the continuity and equicontinuity conditions required by Theorem 3.*

*The same argument applies to convolutional networks, since 2-D convolutions are linear operators whose induced matrix representations also admit spectral norm bounds controlled via spectral normalization.*

## D   Experimental Details

In this appendix, we provide further information on the simulation experiments (Section 6.1) and the real-world wildfire application (Section 6.2), including exact parameter settings, model architecture and execution details, hyperparameter selection strategies, and validation procedures. All code for generating, preprocessing, and analyzing both the synthetic and real-world datasets—and for training and evaluating GST-UNet—is available at `https://github.com/moprescu/GSTUNet`, with step-by-step replication instructions in the repository's `README.md`.

For both applications, GST-UNet employs a U-Net backbone with a single ConvLSTM layer (hidden dimension 32) and a contracting-expanding path of channel sizes $16 \to 32 \to 64 \to 128 \to 256$. The G-computation heads are implemented as shallow feed-forward neural networks that operate on the U-Net feature maps at each grid cell for G-computation. In practice, to ensure stable ConvLSTM training and reduce computational overhead, we truncate the input history to a fixed length. All neural networks are implemented via the `nn` module in `PyTorch` [32]. Experiments were conducted on an NVIDIA A100 (Ampere) GPU using the Perlmutter system at the National Energy Research Scientific Computing Center (NERSC). The synthetic experiments required roughly 55 minutes per hyperparameter set, while the wildfire experiment completed in about 5 minutes.

### D.1   Synthetic Experiments

**Data Simulation Process.** For our primary simulation experiments, we generate $T = 200$ time steps on a $64 \times 64$ grid. The simulation parameters in the generating equations

$$\mathbf{X}_t = \alpha_0 + \alpha_1 \mathbf{X}_{t-1} + \alpha_2 \mathbf{A}_{t-1} + \alpha_3 (K_X * \mathbf{X}_{t-1}) + \epsilon_X,$$

$$\mathbf{A}_t \sim \text{Bern}\Big(\sigma\big(\beta_1\big(\beta_0 + \tfrac{1}{L}\sum_{l=0}^{L-1} K_A * \mathbf{X}_{t-l}\big)\big)\Big),$$

$$\mathbf{Y}_t = \gamma_0 + \gamma_1\big(K_{YA} * \mathbf{A}_{t-1}\big) + \gamma_2 \tfrac{1}{L}\sum_{l=1}^{L}\big(K_{YX} * \mathbf{X}_{t-l}\big) + \gamma_3 \mathbf{Y}_{t-1} + \epsilon_Y,$$

are given by:

- $\mathbf{X}_t$:

$$\alpha_0 = 0.5,\ \alpha_1 = 0.5,\ \alpha_2 = -2.0,\ \alpha_3 = 0.2,\ K_X = \begin{pmatrix} 0 & 0.45 & 0 \\ 0.15 & 0 & 0.35 \\ 0 & 0.05 & 0 \end{pmatrix}.$$

  where $K_X$ influences how $\mathbf{X}$ diffuses across neighboring cells, with an asymmetry due to advection.

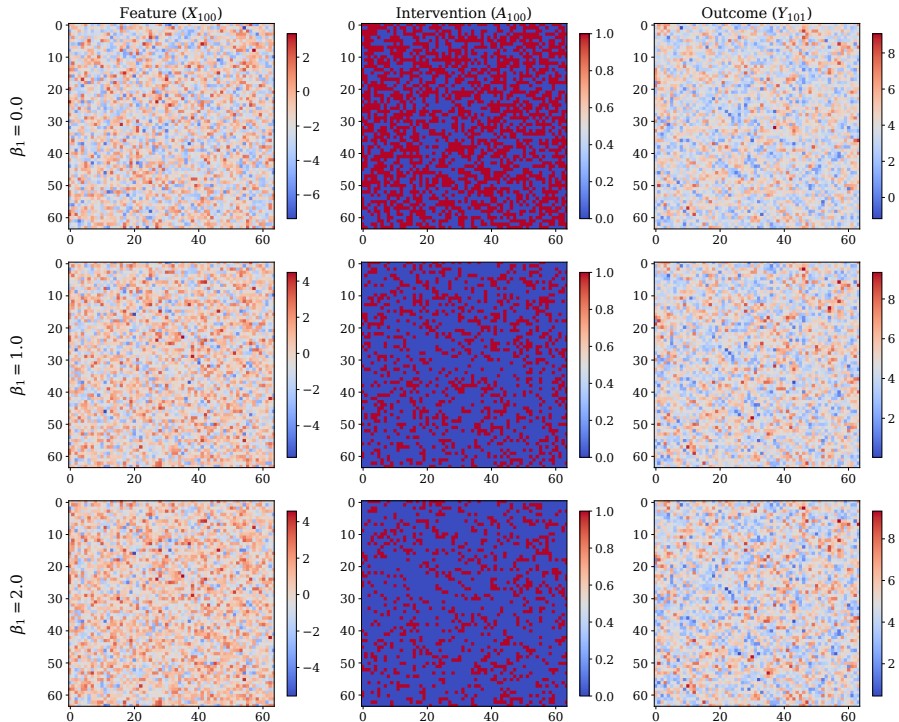

Figure 4: Samples from the DGP at $t = 100$, comparing feature $X_{100}$ (left), intervention $A_{100}$ (center), and outcome $Y_{101}$ (right) for varying $\beta_1 \in \{0.0, 1.0, 2.0\}$.

Table 3: Hyperparameters and their ranges. We boldface the values that provided the best validation performance.

| Hyperparameter | Model(s) | Value Range |
|---|---|---|
| Batch size | All models | $\{2, \mathbf{4}, 8\}$ |
| Learning rate | All models | $\{10^{-4}, \mathbf{5 \times 10^{-4}}, 10^{-3}\}$ |
| Scheduler patience | All models | $\{3, \mathbf{5}, 10\}$ |
| Early stopping patience | All models | $\{5, \mathbf{10}\}$ |
| Curriculum period | GST-UNet | $\{1, \mathbf{3}, 5, 7\}$ |
| Curriculum learning rate | GST-UNet | $\{10^{-4}, \mathbf{5 \times 10^{-4}}, 10^{-3}\}$ |
| UNet output dim $d_h$ | GST-UNet | $\{8, \mathbf{16}, 32\}$ |
| G-head hidden size | GST-UNet | $\{\mathbf{8}, 16\}$ |
| G-head layers | GST-UNet | $\{\mathbf{1}, 2, 3\}$ |

- $\mathbf{A}_t$:

$$\beta_0 = -1.0, \ \beta_1 \in \{0.0, 0.5, 1.0, 1.5, 2.0, 2.5, 3.0\}, \ K_A = \frac{1}{16}\begin{pmatrix} 1.0 & 1.0 & 1.0 \\ 1.0 & 8.0 & 1.0 \\ 1.0 & 1.0 & 1.0 \end{pmatrix}.$$

- $\mathbf{Y}_t$:

$$\gamma_0 = 2.0, \ \gamma_1 = 1.5, \ \gamma_2 = 0.5, \ \gamma_3 = 0.5$$

$$K_{YX} = \frac{1}{16}\begin{pmatrix} 1.0 & 1.0 & 1.0 \\ 1.0 & 8.0 & 1.0 \\ 1.0 & 1.0 & 1.0 \end{pmatrix}, \ K_{YA} = \frac{1}{16}\begin{pmatrix} 1.0 & 1.0 & 1.0 \\ 1.0 & 8.0 & 1.0 \\ 1.0 & 1.0 & 1.0 \end{pmatrix}.$$

We use $L = 5$ temporal lags for $\mathbf{X}$ and $\mathbf{Y}$, a seed of 42 for reproducibility. The parameter values were chosen such that the simulation remains stable (*i.e.*, the process does not diverge). See Figure 4 for representative $t = 100$ snapshots of $X_{100}$, $A_{100}$, and $Y_{101}$ under varying $\beta_1$.

For each $\beta_1$, we first generate a factual dataset of length $T = 200$ (*i.e.*, $\{(\mathbf{X}_t, \mathbf{A}_t, \mathbf{Y}_t)\}_{t=1}^{200}$). We then create $n_{\text{test}} = 50$ test histories of length $l_H = 10$. For each test history, we simulate 100 trajectories

Table 4: Ablation on spatial kernel size ($\tau = 5$). Removing neighbor aggregation ($1 \times 1$ kernel) degrades performance, confirming the need to model spatial spill-overs.

| Kernel size | $\beta_1$=0.0 | 0.5 | 1.0 | 1.5 | 2.0 |
|---|---|---|---|---|---|
| $3 \times 3$ | $\mathbf{0.33 \pm 0.004}$ | $\mathbf{0.35 \pm 0.004}$ | $\mathbf{0.40 \pm 0.005}$ | $\mathbf{0.44 \pm 0.004}$ | $\mathbf{0.40 \pm 0.005}$ |
| $1 \times 1$ | $0.53 \pm 0.004$ | $0.55 \pm 0.005$ | $0.54 \pm 0.005$ | $0.60 \pm 0.007$ | $0.64 \pm 0.006$ |

Table 5: Effect of increasing $T$ on RMSE for $\beta_1 = 2.0$. GST-UNet improves with more data, while baselines remain biased.

| Model | T=100 | T=200 | T=400 | T=600 | T=800 |
|---|---|---|---|---|---|
| UNet+ | 0.78 | 0.81 | 0.82 | 0.95 | 0.87 |
| STCINet | 0.80 | 0.90 | 1.04 | 1.02 | 0.91 |
| GST-UNet | **0.69** | **0.40** | **0.32** | **0.32** | **0.36** |

under a randomly chosen (yet fixed over the test data) counterfactual intervention of length $\tau = 10$, and average the outcomes at each step to approximate the true CAPOs. This procedure yields a final test set of shape $n_{\text{test}} \times (l_H + \tau + 1) \times 64 \times 64$, *i.e.*, $50 \times 21 \times 64 \times 64$.

**Neural Architectures.** The **GST-UNet** comprises a single ConvLSTM layer (hidden dimension 32), followed by a U-Net with channel sizes $16 \rightarrow 32 \rightarrow 64 \rightarrow 128 \rightarrow 256$. Its G-computation heads are shallow feed-forward networks operating on the final U-Net feature maps at each grid cell; both the U-Net's output dimension ($d_h$) and the G-head architecture (number of layers, hidden size) are treated as hyperparameters. The **UNet+** baseline uses the same ConvLSTM+U-Net backbone as GST-UNet but outputs a single channel ($d_h = 1$), omitting any G-computation. For direct comparison, we also implement **STCINet** [1] with an identical ConvLSTM+U-Net backbone, and retaining their original Latent Factor Model (LFM) details.

**IPWUNet Baseline.** We adapt the Inverse Propensity Weighting (IPW) estimator from [49] to the spatiotemporal setting. Given estimated propensities $\hat{\pi}(\mathbf{a}_l \mid \mathbf{H}_{1:l})$, the estimator is defined as:

$$\hat{Y}_{t+\tau}^{\text{IPW}} = \left( \prod_{l=t}^{t+\tau} \frac{\mathbb{I}[\mathbf{A}_l = \mathbf{a}_l]}{\hat{\pi}(\mathbf{a}_l \mid \mathbf{H}_{1:l})} \right), \quad \text{CAPO} = \mathbb{E}[\hat{Y}_{t+\tau}^{\text{IPW}} \mid \mathbf{H}_{1:t} = \mathbf{h}_{1:t}].$$

We implement the IPWUNet baseline by reusing the UNet+ architecture (U-Net + ConvLSTM + Attention) for both propensity estimation and outcome prediction. Specifically, we first train the propensity model with a binary cross-entropy loss to estimate $\hat{\pi}(\mathbf{A}_t \mid \mathbf{H}_t)$ at each time $t$. We then freeze this model and use the estimated weights to train a second instance of the same architecture with a weighted MSE loss, where pseudo-outcomes are reweighted by the estimated inverse propensities along the counterfactual treatment path. While this allows partial adjustment for time-varying confounding, the method does not correct for spatial interference and is sensitive to small propensity values, which can lead to high variance.

**Training Details.** We randomly initialize all model parameters (GST-UNet and baselines) with Xavier uniform weights [17]. We use the Adam optimizer [21] with an initial learning rate, halving it whenever the validation loss plateaus for a specified scheduler patience. To mitigate overfitting, we adopt early stopping when the validation loss fails to improve for a specified early stopping patience epochs. Validation uses 40 of the 190 training prefixes, and the total training is capped at 100 epochs. We tune the following hyperparameters: *(i)* batch size, learning rate, scheduler patience, and early stopping patience (common to all models); *(ii)* for GST-UNet, the curriculum period and learning rate for curriculum phases, the U-Net output dimension $d_h$, and the number and width of hidden layers in the feed-forward G-heads. Table 3 lists the hyperparameter ranges considered, with the values yielding the best validation performance in **bold**.

**Evaluation Procedure.** We evaluate each model by averaging the root mean square error (RMSE) of the estimated CAPOs against ground truth across 50 test trajectories. Table 1 in the main text reports RMSE $\pm$ standard deviation for horizon lengths $\tau \in \{5,\ 10\}$ and $\beta_1 \in \{0, 0.5, 1.0, 1.5, 2.0\}$.

**Effect of Varying $T$.** We ran additional simulations varying the trajectory length $T \in \{100, 200, 400, 600, 800\}$ and $\beta_1 = 2.0$ while keeping the grid size fixed ($d_x = d_y = 64$). Results are shown Appendix D.1. GST-UNet consistently improves with more data, while the baselines

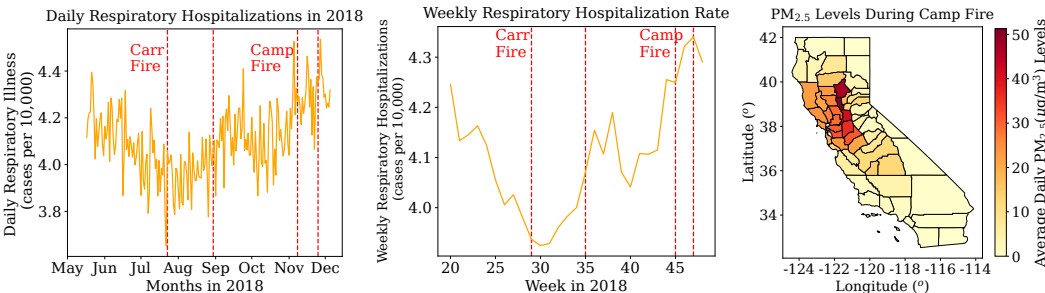

Figure 5: **(Left)** Daily respiratory illness incidence (cases per 10,000). **(Center)** Weekly aggregated incidence. **(Right)** Average daily PM$_{2.5}$ during the Camp Fire.

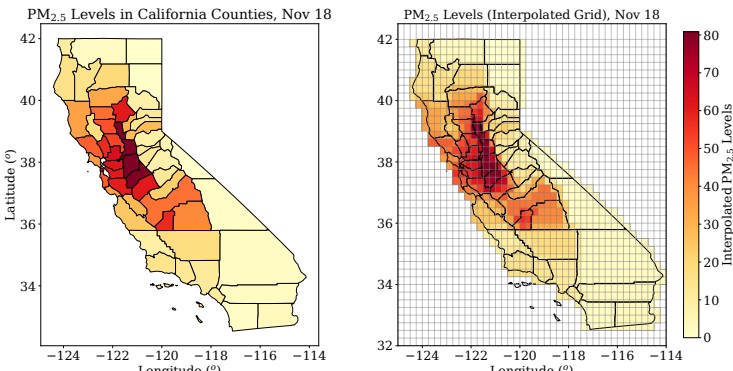

Figure 6: An example of county-level (*left*) vs. grid-interpolated (*right*) PM$_{2.5}$ levels on November 18 (during the Camp Fire). The grid interpolation produces a $40 \times 44$ lattice of area-weighted estimates aligned with our spatiotemporal framework.

remain biased—even as $T$ increases. This highlights the importance of adjusting for time-varying confounding: without it, there is a persistent asymptotic bias.

**Effect of Neighbor Aggregation.** To evaluate the importance of spatial spill-over modeling, we ablate the convolutional kernel used in the ConvLSTM encoder. Table 4 compares GST-UNet with a standard $3 \times 3$ kernel against a variant that removes neighbor aggregation by using a $1 \times 1$ kernel. Across all levels of confounding strength ($\beta_1$), performance deteriorates markedly when neighbor information is excluded, with RMSE increasing by 30–40%. This confirms that explicitly aggregating information from nearby locations is essential for capturing spatial interference and achieving unbiased counterfactual estimates.

## D.2  Wildfire Application

**Data Preprocessing and Interpolation.** We analyze daily, county-level data from Letellier et al. [23] that include PM$_{2.5}$ (particulate matter $< 2.5 \, \mu m$), hospitalization counts for respiratory and cardiovascular conditions, and weather variables (temperature, precipitation, humidity, radiation, wind), plus population estimates from the California Department of Finance. Our study period spans weeks 20–48 (May 18–December 2, 2018), covering both the Carr and Camp fires. As illustrated in Figure 5, daily and weekly aggregated respiratory illness rates rise around these events, while PM$_{2.5}$ levels also surge during the Camp Fire.

To align with our spatiotemporal framework, we use `geopandas` [19] to interpolate county-level covariates, PM$_{2.5}$, and hospitalizations onto a latitude–longitude grid from 32°N to 42°N latitude and -125° to -114° longitude, at a resolution of $0.25°$. Each grid cell's values are an area-weighted average of the counties it intersects, yielding a $40 \times 44$ spatial lattice. We mask out non-California cells by setting them to zero, thus obtaining a consistent dataset for further analysis. Figure 6 illustrates how the raw county-level data compare to the interpolated grid for PM$_{2.5}$ on November 18.

**Model Training and Validation.** We train GST-UNet, UNet+, STCINet, and IPWUNet with a prediction horizon of $\tau = 10$ days. All models use a shared set of hyperparameters: batch size $= 4$,

Table 6: Estimated county-level increases in respiratory ED visits attributable to the wildfire event, with 95% bootstrap confidence intervals. Population is reported in units of 10,000. Counties marked with * have smaller populations, which may lead to greater uncertainty.

| County | Mean | 2.5% | 97.5% | Population ($\times 10^4$) | Interval Width / Population |
|---|---|---|---|---|---|
| Tehama | 37 | -126 | 158 | 6.4 | 44.4 |
| Butte | 168 | 30 | 325 | 23.0 | 12.8 |
| Glenn* | -52 | -262 | 39 | 2.8 | 107.6 |
| Colusa* | 13 | -158 | 107 | 2.1 | 124.0 |
| Sutter | -18 | -170 | 70 | 9.6 | 24.9 |
| Napa | 81 | -41 | 192 | 13.9 | 16.8 |
| Lake | 103 | -66 | 203 | 6.4 | 41.8 |
| Solano | 38 | -79 | 173 | 44.6 | 5.6 |
| Sacramento | 202 | -107 | 484 | 153.9 | 3.8 |

learning rate $= 5 \times 10^{-4}$, scheduler patience $= 5$, early stopping patience $= 10$, and a curriculum period $= 5$ (with curriculum learning rate $= 5 \times 10^{-4}$). For GST-UNet, we additionally set the U-Net output dimension to $d_h = 16$, the G-head hidden layer size to $8$, and the number of G-head layers to $1$. We optimize a mean squared error (MSE) loss with two adjustments: (1) we mask grid cells outside California to exclude them from the loss computation, and (2) we apply cell-specific weights proportional to the number of grid cells per county to avoid bias toward geographically larger counties. Hyperparameter tuning and validation are performed using data from the first 50 days of the wildfire season. Using the selected configuration, we generate counterfactual predictions for the Camp Fire peak period (November 8–17) by iteratively applying each trained model with increasing history lengths. We note that counties with populations below 20,000–30,000 can yield unreliable incidence rate estimates (baseline daily rates of approximately 4 cases per 10,000 individuals); in Figure 3, we denote these high-uncertainty counties with hatched markings.

**Bootstrap Confidence Intervals.** We compute 95% bootstrap confidence intervals for all models using $n = 40$ bootstrap samples, balancing statistical rigor with computational load. Counties with populations below 20,000–30,000 tend to yield unstable incidence rate estimates, driven by low baseline daily counts (approximately 4 cases per 10,000), and are excluded from the analysis. These counties are indicated with hatching in Figure 3, a choice further supported by the bootstrap results. In Table 6, we report bootstrap intervals for the counties closest to the Camp Fire. Glenn and Colusa exhibit disproportionately wide intervals–reflecting the uncertainty introduced by their small population sizes–and this further justifies their exclusion from the final analysis.

# E   Limitations and Broader Impacts

**Limitations.** While GST-UNet demonstrates strong empirical performance and theoretical grounding, several limitations should be acknowledged. First, our method relies on standard causal identification assumptions, including no unobserved confounding (Assumption 1), which is inherently untestable and may not hold in all real-world settings. Second, our framework assumes the existence of a time-invariant representation of the spatiotemporal process (Assumption 2)–a useful but idealized condition that may be violated in domains with highly non-stationary or regime-shifting dynamics. Finally, GST-UNet is designed for gridded spatiotemporal data and assumes a regular spatial lattice; while this is common in environmental and health applications, adapting the framework to irregular spatial structures (e.g., graphs or administrative boundaries) is an important direction for future work.

**Broader Impacts.** This work advances machine learning by introducing a spatiotemporal causal inference framework for estimating treatment effects in complex real-world settings. The GST-UNet has broad applications in public health, environmental science, and social policy, where understanding interventions supports evidence-based decisions. For example, it can inform pollution control, wildfire response, or health resource allocation. However, like all observational methods, GST-UNet depends on the quality and completeness of the data, as well as the assumptions stated in this work. We caution against uncritical use in high-stakes settings, as violations of model assumptions or data biases can lead to misleading conclusions. We encourage responsible deployment–especially in contexts affecting vulnerable populations–and recommend pairing our method with domain expertise, sensitivity checks, and uncertainty quantification.

