# OpenReview forum: "GST-UNet: A Neural Framework for Spatiotemporal Causal Inference with Time-Varying Confounding"
_NeurIPS.cc/2025/Conference — NeurIPS 2025 poster_

### Official Review · Reviewer_RWWK · 2025-06-02

**Clarity:** 3
**Significance:** 2
**Originality:** 1
**Rating:** 2
**Confidence:** 4

**Summary:**

The paper proposes the GST-UNet, a neural method for estimating conditional average potential outcomes (CAPOs) over time. Therein, the authors use an iterative approach to G-computation. Further, they incorporate spatial dependencies in their neural backbone, which is different to existing neural methods. The method is validated on synthetic and real world data.

**Questions:**

Questions:

- What are the key differences to works such as [1] and [2]?
- How can interference between units be accounted for?
- Can the authors add more experimental results against neural baselines that are tailored for CAPO estimation over time?

**Ethical Concerns:**

["NO or VERY MINOR ethics concerns only"]

**Final Justification:**

Thank you for your rebuttal.

The comparison to existing neural methods for estimating CAPOs made -- to some extent -- sense to me, but the novelty still appears quite limited. Further, without experiments showcasing the performance increase due to mis-specified assumptions in the aforementioned baselines, it is difficult to assess the effectiveness of the GST-UNet. For those reasons, I will maintain my score.

**Limitations:**

The authors discuss limitations regarding their assumptions in Appendix E.

**Paper Formatting Concerns:**

no formatting concerns

**Quality:**

2

**Strengths And Weaknesses:**

Strengths:

- The paper is a complete piece of work. Assumptions and proofs are provided, and the theoretical background is well explained.
- The task of interest is highly relevant.

Weaknesses:

- The proposed method is very close to existing works [1,2]. Specifically, [1] uses the exact same iterative regression scheme (generating pseudo-outcomes and learning them in an iterative manner). Further, from a high-level perspective, [1] has the same neural architecture (neural backbone + G-computation heads) and proposes the exact same learning algorithm. Further, [2] has a closely related TMLE approach for estimating ATEs over time, with an additional targeting step to remove first-order bias. At the very least, the authors of the paper under review should acknowledge these existing works where appropriate (e.g., in the architecture, the learning algorithm, etc.) and point out key differences more clearly.
- From my understanding, the GST-UNet only differs from [1] in its neural backbone, which is a U-Net consisting of ConvLSTM layers. The proposed backbone itself, however, is also not novel. Hence, the proposed paper appears to be a simple combination of existing works.
- It remains unclear how dependencies between samples (interference) can be accounted for with the proposed backbone. The only dependencies that are additionally accounted for are spatial, but not the dependencies between units (i=1,...n), which is typically referred to as interference.
- In the experimental section, no neural state-of-the-art methods for CAPO estimation over time are used as baseline (e.g., [1,3,4,5]). Given that one of the main claims is that these baselines cannot handle spatial dependencies, the experimental section seems insufficient.


___

[1] K. Hess, D. Frauen, V. Melnychuk, and S. Feuerriegel. G-transformer for conditional average potential outcome estimation over time. arXiv preprint arXiv:2405.21012, 2024.

[2] Yi Shirakawa, Toru; Li, Yulun Wu, Sky Qiu, Yuxuan Li, Mingduo Zhao, Hiroyasu Iso, and Mark van der Laan. 2024. Longitudinal targeted minimum loss-based estimation with temporal-difference heterogeneous transformer. In ICML.

[3] Valentyn Melnychuk, Dennis Frauen, and Stefan Feuerriegel. Causal transformer for estimating counterfactual outcomes. In ICML, 2022.

[4] Ioana Bica, Ahmed M. Alaa, James Jordon, and Mihaela van der Schaar. Estimating counterfactual treatment outcomes over time through adversarially balanced representations. In ICLR, 2020.

[5] Rui Li, Stephanie Hu, Mingyu Lu, Yuria Utsumi, Prithwish Chakraborty, Daby M. Sow, Piyush Madan, Jun Li, Mohamed Ghalwash, Zach Shahn, and Li-wei Lehman. G-Net: A recurrent network approach to G-computation for counterfactual prediction under a dynamic treatment regime. In ML4H, 2021.

---

> ### Author Rebuttal · Authors · 2025-07-31
>
> **Strengths:**
>
> Thank you for your thorough review. We appreciate that you found the theoretical background clear and the overall task highly relevant. Below we clarify the differences from prior work and address each of your questions; we hope these points will encourage you to reconsider your score.
>
> **Re: Novelty and backbone comparison vs. prior neural G-computation**
>
> Prior neural G-computation methods assume many i.i.d. temporal trajectories, whereas GST-UNet targets a single, spatially coupled sequence where interference, spatial confounding, and data scarcity are first-class challenges. This makes both the assumptions and the required architecture fundamentally different. The table below summarises the key distinctions.
>
> | **Dimension** | **Prior neural G-comp** | **GST-UNet (ours)** |
> |-----|-----|-----|
> | Data structure & assumptions |  *Single spatiotemporal trajectory*—e.g., one 64 × 64 grid set of $(A_{s,t}, X_{s,t}, Y_{s,t})$ observed for $T$ days. Every frame is correlated with its neighbours *in space* (shared weather patterns) and with previous frames *in time* (smoke today affects pollution tomorrow). The target CAPOs themselves are *spatially structured surfaces*—we seek $ \textbf{CAPO}_{s}(a)$ at *every* grid cell, not a single scalar per individual. Because of this tight coupling we cannot break the data into $64 × 64$ independent time-series or assume thousands of separate “videos.” To learn with only $T-\tau$ usable prefix windows, we introduce **Assumption 2 (representation-based time-invariance)**: if a history embedding $\phi(H\_{1:t})$ captures the latent state, then different prefixes behave *as if* they were i.i.d. samples, enabling consistent G-computation in this data-scarce, highly correlated regime. | Independent patient/user time series; no spatial dimension. Outcomes depend only on each unit’s own history, so interference and spatial confounding are out of scope and only temporal relationships are modeled. Identification (when given) relies on i.i.d. sampling across units. |
> |Backbone & learning algorithm | ConvLSTM-UNet encoder couples space + time and leverages learned convolutional kernels that aggregates neighbour treatments/covariates, capturing both direct/spillover effects and spatial confounding. Must be sample-efficient because only $T-\tau$ data prefixes (quasi-i.i.d samples) are available, where $T$ is the length of the spatiotemporal sequence. Curriculum training stabilises the iterative pseudo-outcome recursion, needed due to the complex data structure and scarce data. | Architectures assume arbitrary numbers of i.i.d. trajectories, allowing expressive transformers or RNNs that model *temporal* dependence only. No mechanism for neighbour treatments and covariates; no curriculum for iterative G-comp., most likely due to the simpler nature of the data structure and the larger number of i.i.d. samples |
> | Theoretical results | Unlike previous works (e.g. the STCINet in our experiments), we state the minimally sufficient requirements (Assumption 2) for the identification results (Theorem 1) to hold. We additionally provide consistency of CAPO estimates under the single-trajectory/interference regime (Theorem 2) — new in the G-computation literature.| No prior neural G-comp. work (to our knowledge) provides identification or consistency  guarantees when only one spatiotemporal sequence is observed.|
>
> As the comparison shows, GST-UNet introduces substantial new elements—and considerable care in both design and theory—to deliver an accurate, unbiased, ready-to-use estimator for spatiotemporal causal effect estimation.
>
> **Re: Interference between units**
>
> In our grid, each spatial location is a unit; interference occurs when $A_{s',t}$​ influences $Y_{s,t}$​. The learned convolutional kernel aggregates neighbour treatments and covariates before the G-comp heads, capturing both direct and spillover effects. For the purposes of this rebuttal, we reran the synthetic study with a 1 × 1 kernel (no neighbour information). The results (RMSE ± SD) are given below for horizon $\tau=5$:
>
> | Kernel Size | $\beta_1 = 0.0$ | $\beta_1 = 0.5$ | $\beta_1 = 1.0$ | $\beta_1 = 1.5$ | $\beta_1 = 2.0$ |
> |-------------|------------------|-------------------|-------------------|-------------------|-------------------|
> | \(k = 3\)    | 0.329 ± 0.004     | 0.348 ± 0.004      | 0.402 ± 0.005      | 0.440 ± 0.004      | 0.404 ± 0.005      |
> | \(k = 1\)    | 0.532 ± 0.004     | 0.547 ± 0.005      | 0.537 ± 0.005      | 0.599 ± 0.007      | 0.636 ± 0.006      |
>
> The  RMSE increases when we don’t consider spatial relationships confirm the importance of modelling spatial spillovers. We will include this ablation in the revised manuscript.
>
> **Re: Baselines tailored to CAPO over time**
>
>  We do not benchmark G-Transformer, CausalTransformer, G-Net, etc., because their core assumption—many i.i.d. trajectories with no cross-unit interaction—clashes with our single-trajectory, spatially coupled setting; adapting them would require redesigning their causal graphs and losses, which is beyond this submission’s scope.
>
> We already cite [1–5] in the related-work section and Appendix A; we will add a concise comparison (table above) to the main text for clarity.
>
> **We appreciate your thoughtful feedback and hope the clarifications provided show the value of our contributions and justify a higher score.**

---

### Official Review · Reviewer_zFj6 · 2025-06-23

**Clarity:** 3
**Significance:** 3
**Originality:** 3
**Rating:** 4
**Confidence:** 4

**Summary:**

The paper addresses a significant problem in causal inference: estimating causal effects from spatiotemporal observational data under time-varying confounding, interference, and spatial dependencies.

**Questions:**

What is the time complexity of GST-UNet?

**Ethical Concerns:**

["NO or VERY MINOR ethics concerns only"]

**Final Justification:**

Thank you for addressing my main concerns. I have no further comments on this work and will maintain my rating in support of its acceptance at NeurIPS.

**Limitations:**

Assumptions may be not practical in real-world applications.

**Quality:**

3

**Strengths And Weaknesses:**

Strengths

The proposed GST-UNet framework integrates neural representation learning (U-Net + ConvLSTM + attention) with iterative G-computation for bias-adjusted effect estimation.

It is clearly motivated by practical challenges in public health and environmental policy.

The method builds on sound causal inference principles, including the potential outcomes framework and G-computation.

Weaknesses

The assumption of embedding-based time invariance might be strong in some real-world scenarios.

The comparison methods (e.g., UNet+, IPWUNet) may not be the most state-of-the-art. Why do not compare with causal transformer [1], G-Net [2] and DeepACE [3]?

The experiments conducted on one dimension time-varying confounder, so it is difficult to assess the performance of GST-UNet on multi-dimensions of time-varying confounders.


[1] Melnychuk, Valentyn, Dennis Frauen, and Stefan Feuerriegel. "Causal transformer for estimating counterfactual outcomes." In International conference on machine learning, pp. 15293-15329. PMLR, 2022.

[2] Rui Li, Stephanie Hu, Mingyu Lu, Yuria Utsumi, Prithwish Chakraborty, Daby M Sow, Piyush Madan, Jun Li, Mohamed Ghalwash, Zach Shahn, et al. G-net: a recurrent network approach to g-computation for counterfactual prediction under a dynamic treatment regime. In Machine Learning for Health, pages 282–299. PMLR, 2021.

[3] Dennis Frauen, Tobias Hatt, Valentyn Melnychuk, and Stefan Feuerriegel. Estimating average causal effects from patient trajectories. In Proceedings of the AAAI Conference on Artificial Intelligence, pages 7586–7594, 2023.

---

> ### Author Rebuttal · Authors · 2025-07-31
>
> **Strengths**
>
> Thank you for your positive and encouraging review. We’re glad you appreciated how GST-UNet unites neural spatiotemporal representation learning with iterative G-computation, is grounded in causal-inference principles, and addresses real public-health and environmental-policy challenges. We respond to your questions below and hope these clarifications will prompt you to raise your score.
>
> **Re: Embedding-based time invariance assumption**
>
> Thank you for raising this important point. While Assumption 3 may appear strong, it is substantially weaker than classical stationarity assumptions. Instead of requiring the marginal distributions of $\mathbf{X}_t$​ and $\mathbf{Y}_t$ to be time-invariant, we assume that—once the history $(\textbf{H}\_{1:t}, \text{A}_t)$ is embedded into a latent representation $\phi(\mathbf{H}\_{1:t}, \text{A}_t)$—the conditional distribution of $(\text{X}\_{t+1}, \text{Y}\_{t+1})$ is invariant across time. This reflects the idea that the mechanism generating outcomes is stable once we condition on all relevant confounders, even if the raw distributions evolve over time.
>
> This assumption is motivated by recent work in time-series causal inference (see Assumption 3 in paper), which uses learned embeddings to pool information across time and improve generalization—particularly important in our single-trajectory setting, where time-dependent parameters are hard to estimate reliably. In many applied domains, such as climate, environmental health or epidemiology, transitions conditioned on relevant covariates (e.g., prior exposure, mobility, or weather) often behave consistently, making this a reasonable and practically useful approximation.
>
> We acknowledge that stronger empirical validation of this assumption is an important direction for future work.
>
> **Re: Comparisons with Causal Transformers and related methods**
>
> We intentionally did not include Causal Transformer [1], G-Net [2], or DeepACE [3] because their foundational assumption—many i.i.d. patient-style trajectories with no cross-unit interaction—conflicts with our single, spatially coupled lattice. Adapting them would require (i) redesigning their causal graphs to include neighbour treatments, (ii) revising their loss functions for one trajectory rather than thousands of i.i.d. ones, and (iii) adding spatial inductive bias so they can learn spill-over effects. These steps are non-trivial and beyond this submission’s scope.
>
> We perform a part-for-part comparison between our model and these methods in the table below:
>
> | Dimension                  | Prior neural G-comp methods (e.g. [1–3])                                                                                                                                                                                                 | **GST-UNet (ours)** |
> |---------|---------|------|
> | **Data structure & assumptions** | Independent user/patient time series; no spatial dimension; outcomes depend only on each unit’s own past.  Typical datasets contain tens of thousands of i.i.d. trajectories, so sample size is not a limiting factor.        | Single spatiotemporal chain—one set of $64 \times 64$ lattices $(A_{s,t}, X_{s,t}, Y_{s,t})$ observed for $T$ time steps. After fixing horizon $\tau$, the effective sample size is at most $T - \tau$ quasi-i.i.d. prefixes. Strong space-time coupling breaks down the i.i.d. assumptions and prevents treating each grid cell as its own trajectory. |
> | **Backbone & learning algorithm** | RNN/Transformer models only temporal dependence; no neighbor treatments; no curriculum.                                                                                                                                                 | ConvLSTM-U-Net couples space and time, learns spill-over via a convolutional kernel, and uses curriculum G-computation.                                                                                                                                                                                         |
> | **Theory**                 | No identification/consistency guarantees for one lattice.                                                                                                                                                                                | Assumption 2 + Theorems 1–2 give identification & consistency for a single-trajectory, interference setting.                                                                                                                                                                                                     |
>
> We already cite and compare these works in the related work and Appendix A; we will move a concise comparison (table above) to Appendix A for clarity.
>
> **Re: Higher-dimensional confounders**
>
> Thank you for pointing this out. We extended the synthetic study to include **5 time-varying confounders**. The revised spatiotemporal DGP is:
>
> We index covariate channels by $c \\in \\{1, \\dots, 5\\}$ and let $\bar X_t = \frac{1}{5}\\sum_{c=1}^5 X^{(c)}_{t}$ enote the channel-average field that drives treatment and outcome.
> Spatial convolutions are indicated with “*”.
>
> $$
> \begin{align}
> \mathbf{(Covariates)} \quad X^{(c)}_{t} &= \alpha^{(c)}_0 + \alpha^{(c)}_1 X^{(c)}\_{t-1}
>                + \alpha^{(c)}_2 A\_{t-1}
>                + \alpha^{(c)}_3 (K_X * X^{(c)}\_{t-1})
>                + \varepsilon^{(c)}\_{X,t} \\\\
> \mathbf{(Treatment)}\quad
> A\_{t} &\sim \mathrm{Bernoulli}\bigl(\sigma\bigl(\beta_0 + \beta_1 (K_A * \bar X_t)\bigr)\bigr) \\\\
> \mathbf{(Outcome)}\quad Y_t &= \gamma_0 + \gamma_1 (K\_{YA} * A\_{t-1})
>         + \gamma_2 (K\_{YX} * \bar X\_{t-1})
>         + \gamma_3 Y\_{t-1}
>         + \varepsilon\_{Y,t}
> \end{align}
> $$
>
> We train our method and the corresponding baselines as in the paper and report the results below ($\tau=5$):
>
> | Model     | $\beta_1 = 0.0$   | $\beta_1 = 0.5$   | $\beta_1 = 1.0$   | $\beta_1 = 1.5$   | $\beta_1 = 2.0$   |
> |-----------|---------------------|---------------------|---------------------|---------------------|---------------------|
> | UNet+     | **0.28 ± 0.01**     | 0.39 ± 0.01         | 0.51 ± 0.01         | 0.59 ± 0.01         | 0.65 ± 0.01         |
> | STCINet   | 0.32 ± 0.01         | 0.39 ± 0.01         | 0.50 ± 0.01         | 0.57 ± 0.01         | 0.71 ± 0.01         |
> | IPWUNet   | 0.56 ± 0.01         | 0.59 ± 0.01         | 0.60 ± 0.01         | 0.58 ± 0.01         | 0.65 ± 0.01         |
> | GSTUNet   | 0.37 ± 0.01         | **0.39 ± 0.01**     | **0.44 ± 0.01**     | **0.40 ± 0.01**     | **0.41 ± 0.01**     |
>
> These results confirm GST-UNet’s robustness to increasingly strong time-varying confounding, while plain UNet variants and existing spatial models degrade.
>
> **Re: Time complexity of the GST-UNet**
>
> *Analytical form.* A single forward/back-prop pass scales linearly in
>  $N_{xy}$​ (the number of spatial cells), in the horizon $\tau$ (the number of G-heads), and in the channel width $d$, leading to a runtime that scales roughly like $O(N_{xy} \tau d^2)$.
> *Empirical runtimes* (A100 GPU on NERSC’s Perlmutter cluster) for each experiment:
>
> | Task | Grid & $\tau$ | Time |
> | ----| ---------|--------|
> | Synthetic study | 64 × 64, $\tau$ = 10 | $\simeq 11$ min|
> | Wildfire study | 40x44, $\tau=10$ | $\simeq 5$ min|
>
> **We hope these clarifications are helpful and further strengthen your confidence in the work.**

---

### Official Review · Reviewer_sKpM · 2025-06-26

**Clarity:** 2
**Significance:** 2
**Originality:** 3
**Rating:** 4
**Confidence:** 3

**Summary:**

The paper introduces GST-UNet, an end-to-end neural framework for spatiotemporal causal inference with time-varying confounding. A U-Net + ConvLSTM + attention encoder learns a history-invariant representation, while a stack of \tau “G-heads” implements regression-based iterative G-computation to estimate conditional average potential outcomes (CAPOs). A curriculum schedule gradually activates earlier heads so that pseudo-outcomes become reliable. The authors prove identification (Theorem 1) and consistency (Theorem 2) under standard sequential ignorability plus a weaker “representation-based time-invariance” assumption.

**Questions:**

1. Curriculum schedule sensitivity: How was the “ease coefficient” chosen? Have you tried other schedules, and does performance vary markedly?
2. Head-count vs. horizon: Your implementation fixes the number of G-heads \tau equal to the longest horizon evaluated. What happens if \tau is over- or under-specified? A small ablation (e.g., \tau ∈ {4, 6, 8, 10}) would help increase robustness.
3. Section 5.4 mentions future work on graphs, but GST-UNet currently assumes a regular grid for convolutions and skip connections. Can you outline concretely how you would replace the ConvLSTM encoder with a graph-message-passing block, and whether any of the proofs need modification?
4. Figure 4 reports training time on 64×64 grids; how does memory usage and wall-clock scale for inputs with more grids (e.g., 256×256 or 512×512), and for longer \tau?
5. For deployment in, say, wildfire smoke early-warning, what is the per-timestep inference latency (ms per grid cell) on an A100 and a CPU?
6. Your synthetic DGP uses a Gaussian smoothing kernel for spill-over. Does GST-UNet require the same spill-over kernel at train and test time?
7. Have you inspected the temporal-attention weights or produced saliency maps to verify that the model focuses on plausible meteorological or demographic features?

**Ethical Concerns:**

["NO or VERY MINOR ethics concerns only"]

**Final Justification:**

The author's detailed rebuttal has addressed all of my questions and made me understand the paper better; therefore, I decided to raise my rating from 3 to 4.

**Limitations:**

yes

**Paper Formatting Concerns:**

No major formatting issues in this paper.

**Quality:**

2

**Strengths And Weaknesses:**

Strengths:
1. First architecture that unifies deep spatiotemporal representation learning with iterative G-computation, handling interference, carry-over, and evolving confounders without user-specified exposure mappings.
2. Application to wildfire smoke quantifies health burden with spatial granularity—aligning with epidemiological literature—showing practical utility.

Weaknesses:
1. Results depend on one environmental-health example; an additional domain (e.g., mobility interventions, agriculture) or a cross-region generalisation split would strengthen external validity.
2. GST-UNet currently assumes a regular lattice; the authors mention future work on irregular graphs, but no experiment shows feasibility.
3. Only one "ease coefficient" value is reported; how did you choose this? A hyperparameter sensitivity analysis would be more helpful.
4. While runtimes are modest on A100, memory/latency for larger \tau or higher-resolution grids is not reported.
5. No empirical diagnostics are offered for the representation-invariance or no-unobserved-confounding assumptions.
6. GST-UNet does not visualise which spatial features drive G-head predictions; attention heat-maps or Shapley scores over the grid could help.

---

> ### Author Rebuttal · Authors · 2025-07-31
>
> **Strengths**
>
> Thank you for your insightful  review. We appreciate your recognition of GST-UNet as the first architecture to blend spatiotemporal representation learning with iterative G-computation, and of its balance of theory, simulations, and a real wildfire-health application. Below we address each of your questions and concerns, and we hope these clarifications will highlight the full strength of our contribution and justify a higher score.
>
> **Re: Domain breadth**
>
> We agree that an additional domain would strengthen the paper. In parallel with this rebuttal, we are preparing a second real-world study on how short-wave radiation affects Arctic sea-ice concentration (dataset request in progress, following the setup of [1]) and will include those results in the camera-ready—or sooner, if they finish by the end of the discussion period.
>
> **Re: Regular grid vs. graph extension**
>
> GST-UNet’s encoder is **modular** and thus it can be swapped for a graph–message-passing variant almost by "find-and-replace":
>
> | Grid Component   | Graph Analogue (Irregular Mesh) | Implementation Details|
> |---------------------|-----------------|-------------------|
> | 2-D conv in ConvLSTM gates | Graph convolution (GCN, GraphSAGE, or GAT [2-4]) on the static adjacency $N(v)$                         | Replace every `Conv2d` call with a graph-conv layer from PyTorch Geometric; weight sharing still enforces locality. |
> | ConvLSTM temporal recursion        | Graph-ConvLSTM / Graph-GRU ([5,6])   | Simply swap the 2-D conv inside each gate with the graph conv above; hidden-state shapes remain.|
> |UNet | Graph U-Net hierarchy with top-k pooling/unpooling and skip connections [7] | Use the Graph U-Net pooling operator in [7]; pool ratio mirrors the $2\times$ grid stride, and skip tensors are passed as in the original U-Net.|
>
> Moreover, *Theorems 1 & 2 need no modification:* they rely only on the history embedding $\phi(\mathbf{H}\_{1:t})$ satisfying Assumption 2, not on how $\phi$ is produced. Thus replacing grid convolutions with message passing adapts GST-UNet to irregular networks while leaving the G-computation heads and proofs intact. Full implementation and hyper-parameter tuning of this graph variant is promising future work, but beyond the scope of this first grid-based study.
>
> **Re: Curriculum "ease coefficient" & schedule sensitivity**
>
> As shown in **Table 2 of Appendix D**, we tuned the ease coefficient $e_c$​ over the grid $(1, 3, 5, 7)$ for every GST-UNet application. Across all synthetic and wildfire runs, $e_c = 3$ consistently gave the best validation RMSE. Intuitively,
>
> * **Too small** an interval ($e_c = 1$) activates new heads almost every epoch, forcing them to fit highly noisy pseudo-outcomes before later heads stabilize.
>
> * **Too large** an interval ($e_c \ge 5$) leaves the backbone frozen for many epochs; when it is finally unfrozen, the network overfits to late-stage heads and struggles to reconcile earlier steps, hurting long-horizon accuracy.
>
> The middle value strikes a balance between denoising pseudo-outcomes and keeping the shared encoder adaptable throughout training.
>
> **Re: Head-count ($\tau$) vs. prediction horizon**
> Each G-head $k$ learns the conditional outcome function
>
> $$ Q_k \bigl(h\_{1:t+\tau-k}\bigr) =
> \mathbb{E}\left[Y\_{t+\tau} \mid| \mathbf{A}\_{t+k:t+\tau} = \mathbf{a}\_{k:\tau},  \mathbf{H}\_{1:t+\tau-k} = \mathbf{h}\_{1:t+\tau-k} \right].$$
>
> i.e., the response when exactly $k$ steps remain in the horizon. Consequently, a model intended for horizon length $\tau$ needs exactly $\tau$ heads so that every offset $k=1,\dots,\tau$ is covered. If we were to train with fewer heads than the test horizon, the recursion would halt prematurely; using more heads than necessary would leave the earliest heads untrained and yield arbitrary outputs.
>
> **Re: Time/ Memory profiles for larger grids/ $\tau$**
>
> Memory usage and wall-clock for training with $\tau=5$:
>
> | Grid (H × W)   | Cells   | Peak GPU Memory | Train Time |
> |----------------|---------|------------------|-------------|
> | 64 × 64        | 4,096   | 1.15 GiB         | 1.80 min    |
> | 128 × 128      | 16,384  | 4.43 GiB         | 2.33 min    |
> | 256 × 256      | 65,536  | 17.59 GiB        | 7.59 min    |
>
> A forward/back-prop pass scales as
>  $\mathcal{O}\bigl(N_{xy}\,\tau\,d^{2}\bigr)$ —linear in the number of spatial cells $N_{xy}$​ and the horizon $\tau$ (with a quadratic dependence on channel width $d$). The near-linear growth in both memory and runtime shown above matches this prediction.
> Runs for 512 × 512 grids and for horizons $\tau>10$ are queued; their results will be included in the camera-ready version.
>
> **Re: Diagnostics & interpretability (attention / saliency)**
>
> Great suggestion. GST-UNet does not use temporal self-attention, an attention gate on the U-Net skip connections—so there are no time-axis attention weights to plot. Instead, we will be generating (i) Grad-CAM saliency maps to highlight which grid cells influence each G-head and (ii) visualisations of the learned spill-over kernel to show how neighbour treatments affect outcomes. These interpretability plots are underway and will be included in the camera-ready (or sooner if ready during the discussion).
>
> **Re: Inference-time latency on GPU and CPU**
>
> In the wildfire simulation, the inference time is 0.0041 ms/ grid cell with the A100 GPU and 0.011 ms/ grid cell with Perlmutter’s AMD EPYC CPU.
>
> **Re: Spill-over kernel mismatch**
>
> **Clarification.** Our synthetic DGP uses a weighted mean of the neighbors, given by an asymmetric advection–diffusion convolution kernel, not a Gaussian smoother.
>
> **Does GST-UNet need the same kernel size?** No. GST-UNet learns whatever spill-over pattern exists in the data: A small 3 × 3 kernel in the first ConvLSTM, plus the U-Net’s down/upsampling, gives a large effective receptive field.
>
> **We appreciate your thoughtful feedback and hope the clarifications provided show the value of our contributions and justify a higher score.**
>
> [1] S. Ali et al. Estimating direct and indirect causal effects of spatiotemporal interventions with spatial interference. ECML PKDD, 2024.
>
> [2] T. N. Kipf and M. Welling. Semi-supervised classification with graph convolutional networks. arXiv preprint arXiv:1609.02907, 2016.
>
> [3] W. Hamilton, Z. Ying, and J. Leskovec. Inductive representation learning on large graphs. NeurIPS, 2017.
>
> [4] P. Veličković, et al. Graph attention networks. arXiv preprint arXiv:1710.10903, 2017.
>
> [5] Z. Cui et al. Traffic graph convolutional recurrent neural network: A deep learning framework for network-scale traffic learning and forecasting. IEEE Trans. Intell. Transp. Syst., 2019.
>
> [6] K. K. Bhaumik et al. STLGRU: Spatio-temporal lightweight graph GRU for traffic flow prediction. PAKDD, Springer, 2024.
>
> [7] H. Gao and S. Ji. Graph U-Nets. ICML, 2019.

---

> ### Comment · Reviewer_sKpM · 2025-08-03
>
> Thank you very much for the detailed response, which has addressed most of my questions and concerns. I am willing to update my score accordingly.

---

> > ### Author Response · Authors · 2025-08-05
> > **Happy to Continue Discussion**
> >
> > Thank you for your reply and for considering our clarifications, we appreciate your engagement with the paper. Let us know if there are any remaining concerns during the discussion period.

---

### Official Review · Reviewer_XYsK · 2025-06-30

**Clarity:** 3
**Significance:** 3
**Originality:** 1
**Rating:** 4
**Confidence:** 3

**Summary:**

The paper proposes the GST-UNet, a neural method for estimating conditional average potential outcomes over time. Therein, the authors combine interative G-computation with a neural backbone to model spatial-temporal dependencies, which is different to existing neural methods. The method is validated on synthetic and real world data.

**Questions:**

Can the authors ellaborate more clearly on their methodological contribution as compared to existing works? Concrete examples, such as specific changes in architecture, learning algorithm, theory etc would be helpful. I am happy to increase my score if the authors can provide convincing arguments.

**Ethical Concerns:**

["NO or VERY MINOR ethics concerns only"]

**Final Justification:**

I believe this is a borderline decision (see my response to the authors for details).

**Limitations:**

Well addressed in Appendix E.

**Paper Formatting Concerns:**

No concerns here.

**Quality:**

2

**Strengths And Weaknesses:**

Strengths:

- The paper is a complete piece of work. Assumptions and proofs are provided, and the theoretical background is well explained.
- The task of interest is challenging and relevant in a variety of applications, including personalized medicine.
- The paper is well written and the complex methodology is well illustrated.

Weaknesses:

- My main point of concern is the novelty of the proposed novelty as compared to existing work. The proposed method is very close to existing works [1, 2, 3, 4] that leverage G-computation within neural architecture. Specifically, [3, 4] seems to use the exact same iterative regression scheme (generating pseudo-outcomes and learning them in an iterative manner to estimate CAPO/ CATE). Further, from a high-level perspective, [3] seems to have a similar neural architecture (neural backbone + G-computation heads) and proposes a very similar learning algorithm. [1, 2] leverage very similar approaches for estimating ATEs over time, with an additional targeting step to remove first-order bias. At the very least, the authors of the paper under review should acknowledge these existing works where appropriate (e.g., in the architecture, the learning algorithm, etc.) and point out key differences more clearly.
- From my understanding, the GST-UNet only differs from [1] in its neural backbone, which is a U-Net consisting of ConvLSTM layers. The proposed backbone itself, however, is also not novel. Hence, the proposed paper appears to be a simple combination of existing works.
- It remains unclear how dependencies between samples (interference) can be accounted for with the proposed backbone. The only dependencies that are additionally accounted for are spatial, but not the dependencies between units (i=1,...n), which is typically referred to as interference.
- In the experimental section, no neural state-of-the-art methods for CAPO estimation over time are used as baseline (e.g., [3, 4, 5, 6]), which seems insufficient since one of the main claims is that these baselines cannot handle spatial dependencies.


References

[1] Frauen et al. Estimating average causal effects from patient trajectories.
[2] Shirakawa et al. Longitudinal targeted minimum loss-based estimation with temporal-difference heterogeneous transformer.
[3] Hess et al. G-Transformer for Conditional Average Potential Outcome Estimation over Time.
[4] Xiong et al. G-Transformer: Counterfactual Outcome Prediction under Dynamic and Time-varying Treatment Regimes.
[5] Lim et al. Forecasting Treatment Responses Over Time Using Recurrent Marginal Structural Networks.
[6] Bica et al. Estimating Counterfactual Treatment Outcomes over Time Through Adversarially Balanced Representations.

---

> ### Author Rebuttal · Authors · 2025-07-31
>
> **Strengths**
>
> Thank you for your thoughtful review. We’re glad you found the theoretical exposition clear, the methodology well illustrated, and the problem we tackle broadly relevant. We address your concerns below and hope these clarifications will prompt a reconsideration of your score.
>
> **Re: Novelty and backbone comparison vs. prior neural G-computation work**
>
> Thank you for bringing this up. In short, prior neural G-computation models tackle many independent, purely temporal trajectories, whereas GST-UNet targets a single, spatially coupled sequence in which interference, spatial confounding, and data scarcity are first-class challenges—so both the assumptions and the required architecture are fundamentally different. We include a side-by-side comparison with previous CAPO-estimation works in the table below:
>
> | **Dimension** | **Prior neural G-comp** | **GST-UNet (ours)** |
> |-----|-----|-----|
> | Data structure & assumptions |  *Single spatiotemporal trajectory*—e.g., one 64 × 64 grid set of $(A_{s,t}, X_{s,t}, Y_{s,t})$ observed for $T$ days. Every frame is correlated with its neighbours *in space* (shared weather patterns) and with previous frames *in time* (smoke today affects pollution tomorrow). The target CAPOs themselves are *spatially structured surfaces*—we seek $ \textbf{CAPO}_{s}(a)$ at *every* grid cell, not a single scalar per individual. Because of this tight coupling we cannot break the data into $64 × 64$ independent time-series or assume thousands of separate “videos.” To learn with only $T-\tau$ usable prefix windows, we introduce **Assumption 2 (representation-based time-invariance)**: if a history embedding $\phi({H}\_{1:t})$ captures the latent state, then different prefixes behave *as if* they were i.i.d. samples, enabling consistent G-computation in this data-scarce, highly correlated regime. | Independent patient/user time series; no spatial dimension. Outcomes depend only on each unit’s own history, so interference and spatial confounding are out of scope and only temporal relationships are modeled. Identification (when given) relies on i.i.d. sampling across units. |
> |Backbone & learning algorithm | ConvLSTM-UNet encoder couples space + time and leverages learned convolutional kernels that aggregates neighbour treatments/covariates, capturing both direct/spillover effects and spatial confounding. Must be sample-efficient because only $T-\tau$ data prefixes (quasi-i.i.d samples) are available, where $T$ is the length of the spatiotemporal sequence. Curriculum training stabilises the iterative pseudo-outcome recursion, needed due to the complex data structure and scarce data. | Architectures assume arbitrary numbers of i.i.d. trajectories, allowing expressive transformers or RNNs that model *temporal* dependence only. No mechanism for neighbour treatments and covariates; no curriculum for iterative G-comp., most likely due to the simpler nature of the data structure and the larger number of i.i.d. samples |
> | Theoretical results | Unlike previous works (e.g. the STCINet in our experiments), we state the minimally sufficient requirements (Assumption 2) for the identification results (Theorem 1) to hold. We additionally provide consistency of CAPO estimates under the single-trajectory/interference regime (Theorem 2) — new in the G-computation literature.| No prior neural G-comp. work (to our knowledge) provides identification or consistency  guarantees when only one spatiotemporal sequence is observed.|
>
> As the comparison makes clear, GST-UNet introduces substantially novel elements—and considerable care in both design and theory—to deliver an accurate, unbiased, and ready-to-use estimator for **spatiotemporal** causal effect estimation.
>
> Finally, we do not benchmark against G-Transformer, CausalTransformer, or other temporal CAPO baselines because their core assumption—many i.i.d. trajectories with no cross-unit interaction—fundamentally clashes with our single-trajectory, spatially coupled setting; adapting those methods would entail redesigning their causal graphs and losses, which lies beyond the scope of this submission.
>
> While we already mention works [1-6] in the related work section and Appendix A, we will add a concise comparison like this table to the final version of the paper.
>
> **Re: Interference**
>
> In our grid, each spatial location **is** a unit; interference arises when $A_{s',t}$​ influences $Y_{s,t}$​. The learned convolutional kernel aggregates neighbour treatments/covariates before the G-comp heads, capturing both direct and spillover effects.  For further validation, we reran the synthetic study with a 1 × 1 kernel (no neighbour information). The results (RMSE ± SD) are given below for horizon $\tau = 5$:
>
> | Kernel Size | $\beta_1 = 0.0$ | $\beta_1 = 0.5$ | $\beta_1 = 1.0$ | $\beta_1 = 1.5$ | $\beta_1 = 2.0$ |
> |-------------|------------------|-------------------|-------------------|-------------------|-------------------|
> | \(k = 3\)    | 0.329 ± 0.004     | 0.348 ± 0.004      | 0.402 ± 0.005      | 0.440 ± 0.004      | 0.404 ± 0.005      |
> | \(k = 1\)    | 0.532 ± 0.004     | 0.547 ± 0.005      | 0.537 ± 0.005      | 0.599 ± 0.007      | 0.636 ± 0.006      |
>
> The  RMSE increases when we don’t consider spatial relationships confirm the importance of modelling spatial spillovers. We will include this ablation in the revised manuscript.
>
> **We hope our responses have addressed your key concerns and would be grateful if you would consider a higher score in light of these clarifications.**

---

> > ### Comment · Reviewer_XYsK · 2025-08-03
> >
> > Thank you for your rebuttal, which clarifies some of my concerns regarding the difference in settings and how interference comes into play. I have increased my score to 3, and I remain open to further discussion.
> >
> > I have some remaining concerns regarding the novelty of the proposed method. I understand the difference in settings, but it is not quite clear to me how this precisely translates to substantial differences in methodology.
> >
> > My key question is: *What exactly is so different in the single (spatial) time series setting that requires substantial changes in methodology, beyond architecture adaptations?*
> >
> > Points to consider:
> >
> > - I understand that one central selling point is the novel architecture; however, I am not quite sure whether this alone is a sufficient contribution to warrant acceptance, given the great similarity to previous neural G-computation approaches in terms of the learning algorithm.
> > - Learning from $T-\tau$ prefix windows has been done in previous work, e.g., Causal Transformer [1] to increase sample efficiency even when multiple i.i.d. trajectories are available. Even though these works do not provide identifiability justification for this approach (which I appreciate here), they are very much applicable as baselines, and the method itself does not seem novel.
> > - While some of the assumptions/theories are slightly different in the single time series setting, the identification result (G-formula) is identical to the classical one for the i.i.d. setting. It would be helpful if you could clarify if the novel identification result requires substantial theoretical innovation, and if yes, in what part of the proof. Additionally, to the best of my knowledge, learning from a single time series via prefix windows has been theoretically analyzed, even though in slightly different settings (e.g., [2]). It would be helpful if you could clarify your contributions here.
> >
> > References
> >
> > [1] Melnychuk et al. Causal Transformer for predicting counterfactual outcomes.
> >
> > [2] Kallus et al. Efficiently Breaking the Curse of Horizon in Off-Policy Evaluation with Double Reinforcement Learning.

---

> ### Author Response · Authors · 2025-08-05
> **Clarifying Methodological Contributions Beyond Architecture**
>
> Thank you for the follow-up and for raising these points, we appreciate the chance to clarify. Below, we expand on your remaining questions regarding novelty, methodological contributions, and theoretical results.
>
> **Re: Why the single-trajectory, spatial setting forces methodological changes** (beyond simply swapping the backbone)
>
> Main differences, side-by-side:
>
> | Aspect                                 | Prior neural G-comp (many i.i.d. 1-D trajectories)                              | GST-UNet (one 3-D spatio-temporal chain)                                                                                                                                                                                            |
> |----------------------------------------|----------------------------------|-------------------------------------------------------------|
> | **Training samples**                   | Total windows = **$N_\text{traj} \times (T - \tau)$**; each window is independent. | Total windows = **$ T - \tau$** only. We harvest *every* prefix **and** treat the $H \times W$ grids inside each prefix as additional supervision targets, but they are *correlated* and must be processed jointly. |
> | **Input representation**               | A prefix is a length-$\tau$**vector** $\in \mathbb{R}^{d\tau}$.         | A prefix is a length-$\tau$ **tensor** $\in \mathbb{R}^{d\tau \times H \times W}$ with strong spatial correlation. Requires a joint encoder (ConvLSTM-UNet) that keeps locality.                                            |
> | **Interference / spatial confounding** | Not modelled; loss decomposes over units.                                       | **Learned 2-D kernels** aggregate neighbour covariates + treatments *before each G-head*, capturing spill-over and spatial confounding.                                                                                             |
> | **Optimisation for small $T - \tau$**    | Standard end-to-end training; abundant data.                                    | **Curriculum training:** unlock G-heads from short to long horizon so early pseudo-outcomes stabilise; critical when only $T - \tau$ windows exist.                                                                             |
> | **Sample-efficiency lever**            | Relies on data volume plus generic regularisers.                                | Leverages **weight sharing across grids** (translation invariance) + shared spill-over kernels to turn $H \times W$ correlated targets into an effective data multiplier while respecting dependence.                          |
> | **Theoretical requirements**           | Identification assumes i.i.d. units (standard G-formula).                       | **Assumption 2** (representation-based time-invariance) makes prefix windows *quasi-i.i.d.* $\Rightarrow$ nested expectations hold for a *single* trajectory; followed by a new consistency proof (Thm 2) for the curriculum-trained estimator. |
>
> **Key take-away:** in the single-trajectory setting we must (i) reshape the raw data, (ii) couple losses across space, (iii) stabilise training with a curriculum, and (iv) prove identification/consistency under time-invariant representations—changes that go well beyond swapping a backbone.
>
> **Re: Why Causal Transformer-style baselines are not applicable here**
>
> * **Data volume.** Causal Transformer consumes $N_\text{traj} (T - \tau)$ independent tokens; we have only $T - \tau$ correlated tokens.
>
> * **Spatial dimension.** Its input is a length-$\tau$ vector per unit; extending it to $(H \times W)$ grids would require
>  (i) stacking 4,096 separate “units” for a $64 \times 64$ grid,
>  (ii) giving the model a bespoke exposure kernel to mix them, and
>  (iii) rewriting the loss to couple those units inside each prefix.
>
> * **Interference.** Without these modifications, the baseline would treat every grid cell’s history independently and cannot learn spill-over effects. Thus, any performance gap would reflect a mismatch of assumptions, not algorithmic merit.
>
> These extensions are non-trivial and out of scope for our current work.
>
> **Re: What's novel about the identification result?**
>
> While our G-formula is structurally similar to the classical i.i.d. case, its validity in our setting relies on **Assumption 2**, which ensures that conditioning on a learned embedding renders prefixes quasi-i.i.d.. This is what allows us to treat each prefix as a valid unit for regression, even though they come from a single dependent trajectory with interference (loosely analogous to martingale CLTs, where conditional independence is required given past information).
>
> **Please let us know if there is anything else we can clarify or expand on during the discussion period to help support your evaluation.**

---

> > ### Comment · Reviewer_XYsK · 2025-08-08
> >
> > Thank you for your response and clarifications. I feel like I now have a better understanding of the strengths and weaknesses of the paper, which are, in my opinion:
> >
> > + Rigorous identification results in the spatiotemporal setting
> > + Architecture and training heuristics which seem to work well empirically
> > + Interesting setting and application
> >
> > - Methodological novelty. I still think the novelty is relatively limited and reduces to combining existing architecture adaptations and training heuristics with (also existing) neural G-computation
> > - Baselines/ experiments. Usually, I am not a strong proponent of exhaustive benchmarking in causal inference, as many works are theoretically grounded and can only be benchmarked appropriately using synthetic data. However, since the contribution is mainly on the model architecture/training side (and somewhat heuristic), I do think that a rigorous empirical evaluation is essential. I understand your argument that existing baselines for the i.i.d. setting will most likely be worse as they ignore spatial correlation and interference. Nevertheless, I do think showing this is important (sometimes even the worst-looking baselines can turn out to be quite strong).
> >
> > I believe this is a borderline decision and I do neither hold a firm stance towards acceptance nor rejection. Given the clarifications, I will increase my score to 4 and discuss with the other reviewers. In case of resubmission, I would encourage you to add the missing i.i.d. baselines to improve the paper.

---

> > > ### Author Response · Authors · 2025-08-08
> > >
> > > Thank you for your thoughtful follow-up and willingness to reconsider after our clarifications. We really appreciate it! Regardless of the outcome, message received—we will add the i.i.d. baselines as comparisons in future versions to clearly show the performance gap.

---

### Official Review · Reviewer_GdrS · 2025-07-02

**Clarity:** 2
**Significance:** 3
**Originality:** 3
**Rating:** 4
**Confidence:** 3

**Summary:**

The paper introduces GST-UNet, an end-to-end framework for estimating potential outcomes and treatment effects in spatiotemporal settings that exhibit interference, spatial confounding, temporal carry-over, and time-varying confounding. GST-UNet uses a U-Net and a ConvLSTM to learn spatiotemporal representations, and a set of G-heads that implement G-computation to adjust for confounders. A multi-task loss lets all heads and the encoder train stably. The authors provide a proof of identification (Theorem 1) under standard assumptions plus a representation-based time-invariance assumption, and they also provide a proof consistency in the appendix (theorem 2). Experiments on synthetic data show superior performance of GST-UNet compared to the baselines used in the paper. Additionally, a case study on the health outcomes of wildfire further demonstrates the utility of the model on an empirical example.

**Questions:**

Q1. Generally, the weaknesses listed each pose a question, but I would like to repeat the question referring to w1: What is exactly the basis of the claim on the method being suitable for data-scarce regimes?

Q2. Do you think a network causal inference method could be one to compare against, given that the spatial information can be modeled by modeling the input graph of the treatment unit as a grid (please correct me if I’m missing something)?

**Ethical Concerns:**

["NO or VERY MINOR ethics concerns only"]

**Final Justification:**

As I explained in my initial review and my comment on the rebuttal, I acknowledge the paper's merits in addressing an interesting and relevant problem through a suitable approach and methodology, and with novel contributions. I initially had concerns regarding some of the claims (e.g., why this is suitable for data-scarce regime) and how the contribution was presented (e.g., a main theoretical result being in the appendix, while the main paper thoroughly discussed details of the heuristics). I also believe the initial submission jumps back and forth between discussing the main concepts and presenting conceptual and theoretical arguments, versus extensive implementation details (which again, involve several heuristic choices). The authors' rebuttal however seems convincing to me and their revision plan is reasonable. Of course, to be fully confident about whether the issues will be resolved in the camera-ready version, one needs to see the revised paper, but I believe that is not how this review process works and we need to present an assessment according to the information in hand (that is also the fair way to assess the authors' work; they clearly have put a substantial amount of effort into preparing a convincing rebuttal and a thoughtful revision plan).

TL;DR: I had some concerns and questions, which initially brought me to recommend a weak reject despite recognizing the motivation and the merits of the paper. The rebuttal answered my questions. The authors' promised revision is critical for accepting the paper, but I find their plan reasonable and convincing. In conclusion, I am raising my score to an accept score.

**Limitations:**

This is not the shortcoming of the paper, but similar to essentially all causal inference methods, lack of real-world data for evaluation is a significant limitation, which makes a stronger theory or careful experimental design more important.

**Paper Formatting Concerns:**

No major formatting concerns

**Quality:**

3

**Strengths And Weaknesses:**

**Strengths:**

- The authors properly identify an important gap in representation learning for spatio-temporal  causal inference, and the paper takes important steps in addressing this.
- There is significant effort put into a careful design of the framework, to address this gap. The paper shows significant potentials.
- The paper shows theoretical results, careful methodology, and experiments, including a simulation for testing the models (given the natural limitations of empirical validation in causal inference).
- The authors utilize a well-suited training strategy using curriculum learning to overcome challenges in training G-computation heads simultaneously on noisy recursively-generated targets.

---

**Major Weaknesses:**

To my understanding the main shortcoming of the paper is due to w3 and w4 below, which make it difficult to set clear expectations and settle on a criteria to assess whether the paper is ready to be published or needs substantial improvement (hence a later resubmission).

w1. The paper claims the proposed method is particularly suitable for data-scarce regimes, but I find it hard to see an appropriate discussion of that in the main paper. Are the authors claiming this is implied from their model architecture (e.g. using UNet)? If so, I don’t think that is enough to emphasize on the suitability of the method for data-scarce regimes.

w2. Why is theorem 2 in the appendix? I think the main paper contains quite a lot of unnecessary details that could be deferred to the appendix and replaced with theorem 2, especially given that the authors claim that theorem 1 and 2 are the main theoretical contributions (identification and consistency).

w3. Is the paper primarily proposing a rather general framework supported by promising theory that could serve as a first step for developing spatio-temporal causal inference tools, or are they proposing the tool itself? The paper seems to swing between the two. I doubt that claiming both is a suitable strategy, because in that case the results are doubly insufficient.

w4. Related to w3:
 - a. I find the paper’s presentation unclear for a theoretical paper that proposes a method based on the theory as a stepping stone for future tools to advance spatio-temporal causal inference.
 - b. I also find the experiments insufficient for a paper that proposes a ready to use tool. There are several heuristic/engineering choices in the design of the framework, which need a more thorough and better designed set of experiments to evaluate. e.g., one expects an ablation study, or at least experiments that evaluate how would alternative choices (e.g., replacing UNet with an alternative) impacts the performance.

As I was reading the paper, I initially got the impression that the paper is of the former type (theory-inclined), but later on my impression started to swing towards the latter (ready to use tool) as I read through extensive implementation details (overly elaborate for a theoretical paper), did not find theorem 2 in the main paper, and the elaborate discussion of the experiments.


---

**Minor Weaknesses:**

m1. I found the presentation somewhat confusing. The paper jumps between main concepts and remarks or delves into details so frequently that it becomes distracting from the main contributions and choices. Ideally, it should be clear to the reader which choices are fundamental and of conceptual importance, and which ones can be replaced with an acceptable alternative.

m2. I am not familiar with the literature on spatio-temporal causal inference, so I cannot tell if there are other benchmarks or datasets or not. If there are, it would substantially improve the paper to include them. Otherwise, perhaps a more careful discussion of the simulation (why this specific data generating process, what if a different one is used, etc.) and ideally experiments on more simulations could clarify the utility of the proposed method.

---

**Note:** I believe the paper shows significant merits and potentials (see the strengths). I am just not yet convinced that the current draft is ready to properly show and advocate for these merits, due to the reasons discussed above.

---

> ### Author Rebuttal · Authors · 2025-07-31
>
> **Strengths**
>
> Thank you for your detailed and constructive review and for highlighting our work addressing a key gap in spatiotemporal causal inference and GST-UNet’s thoughtful design, training, and balance of theory with simulations. We respond to your points below and hope these clarifications will show the full strength of our work and earn a higher score.
>
> **Re w1: "Data-scarce regimes"**
>
> Thank you for raising this point. GST-UNet operates in an intrinsically data-scarce setting:
>
> * **Single spatiotemporal chain.** We observe only one evolving lattice $(A_{s,t},X_{s,t},Y_{s,t})_{t=1}^T$​. After choosing a prediction horizon $\tau$, the effective sample size is at most $T-\tau$ quasi-i.i.d. prefixes. To get a sense of scale differences, non-spatial CAPO methods — usually with a Transformer backbone — can use arbitrarily many (usually tens of thousands) independent time-series (e.g. data from different patients). Space–time interactions preclude treating each grid cell as an i.i.d. series.
>
> * **Inductive-bias-driven efficiency.** With so few samples, performance hinges on a backbone whose built-in biases match the physics:
>
>     * The ConvLSTM-U-Net shares weights across location and scale, enforcing locality and translation invariance—exactly the properties needed for atmospheric fields for example. Consequently, U-Net-style backbones have become the de facto choice in earth science modelling and related spatial sciences (e.g., [1, 2, 3, 4, 5] below).
>
>    * Architectures with weaker locality priors (e.g., vanilla visual transformers with global attention [3]) require substantially more data or pre-training to learn comparable patterns, as documented in the original ViT literature; preliminary tests in our setting confirm this trend. We are running a ViT ablation and will include the results in the final version.
>
> * **Backbone agnostic in principle.** Section 4 states that any encoder producing embeddings satisfying Assumption 2 will inherit Theorems 1 & 2. U-Net is simply a strong default for limited data; practitioners can swap in graph-convs or hybrid CNN-ViT blocks as larger datasets become available.
>
> We will add a short paragraph in Section 4 summarising the argument above and include the ViT ablation to make the data-scarce claim explicit.
>
> **Re w2: Theorem 2**
>
> Thank you for flagging this. Theorem 2 was placed in the appendix purely for space, not because it is secondary. In the camera-ready we will: 1) Move Theorem 2 (consistency) into the main text alongside Theorem 1, and 2) Shift some verbose setup, architectural exposition and training-schedule details to the appendix to stay within the page limit.
> We note that Theorem 2 is, to our knowledge, the first consistency result for CAPO estimation with iterative G-computation, so we agree that highlighting it in the main body will make the paper’s theoretical contribution clearer.
>
> **Re w3: Framework vs tool**
>
> Our goal is both to close a *theory gap* and to deliver a *practical recipe*—but the two pieces are tightly coupled, not competing claims.
>
> * **Theory gap.** Outside of STCI-Net [2] (which ignores time-varying confounding and offers no identification guarantees) there is no work that tells practitioners *when* CAPOs can be recovered from a single, spatially coupled chain that is common in Earth science applications, epidemiology, etc. Assumption 2 (representation-based time invariance), Theorem 1 (identification) and Theorem 2 (consistency) fill that gap by stating minimal conditions under which unbiased estimation is even possible. We note that Assumption 2 subsumes (and thus relaxes) the strict stationarity/mixing conditions usually imposed in single-series time-series work, and Theorem 2 is, to our knowledge, the first consistency result for CAPO‐over‐time estimation.
>
> * **Methodology gap.** Once those conditions are known, a practitioner still needs a concrete procedure that (i) respects the single-trajectory sample constraint, (ii) reliably accounts for spatial effects (interference, confounding), and (iii) adjusts for time-varying confounding via iterative G-computation. GST-UNet is one ready-to-use instantiation of the general framework: ConvLSTM-UNet + curriculum-stabilised G-computation heads. We emphasise in Section 4 that *any* encoder satisfying Assumption 2 can replace the ConvLSTM-UNet—our code merely provides a non-trivial, field-tested starting point.
>
> Thus the paper is positioned as *theory-guided tool*: theory tells the practitioner *what must hold* and *how iterative G-computation should be structured*; GST-UNet shows *one concrete way* to achieve it today, while providing a blueprint for future backbones (e.g., graph U-Net and graph convs for irregular meshes).
>
> **Re w4 & m1: Presentation and Experiments**
>
> Point well taken re the presentation. Thus, we propose to reorganize the camera-ready version as follows:
>
> * Promote Theorem 2 (consistency) to the main text and move low-level implementation details to the appendix
>
> * Add a concise roadmap that tags each element as a *core principle* or a *replaceable implementation/engineering choice*, preventing concept–detail jumps.
>
> While the current paper already ablates curriculum training and the attention mechanism, we commit to strengthen the evidence for the current implementation by:
>
> * adding a TinyViT [7] backbone ablation to test robustness to encoder choice, and
>
> * including a second real-world study on the impact of short-wave radiation on Arctic sea-ice concentration (dataset request under way, following the setup of [2]).
>
> We believe that these additions will reinforce the “ready-to-use tool” contribution that the reviewer expects. We hope the streamlined presentation and the new empirical evidence would address your concerns and make the dual contribution—rigorous foundation and practitioner-friendly recipe—clearer.
>
> **Re m2 & Q2: Benchmarks & Network causal inference**
>
> Our synthetic study emulates pollutant advection–diffusion on a 2-D lattice:
>
> * **Physical realism.** Each covariate field $X_{s,t}$ evolves via a discretized advection–diffusion equation with an **asymmetric kernel** $K_X$ that mimics wind-driven transport; treatments $A_{s,t}$​ inject additional emissions that propagate through the same kernel. This produces both temporal carry-over and spatial spillover—the two phenomena GST-UNet is designed to handle.
>
> * **Time-varying confounding.** Past emissions alter local $tX_{s,t}$​, which in turn influence future $A_{s,t}$ choices, mirroring real regulatory feedback loops.
>
> * **Why this DGP?** It is (i) analytically tractable, (ii) tunable for different spillover radii, and (iii) aligned with our real-world wildfire application, allowing direct “toy-to-real” translation.
> Thus, while simplified, the setup is grounded in a real physical process.
>
> At present, **no public semi-synthetic benchmark** combines both interference and time-varying confounding. Prior spatial works such as Huang et al. [8] model only static spatial confounding and explicitly list dynamic extensions as future work; contemporaneous CAPO papers [2, 9, 10] likewise rely on fully synthetic or domain-specific real data. Constructing a comprehensive benchmark is a valuable but separate research effort that we plan to undertake in the future.
>
> Additionally, most network-causal methods (see e.g. [11-13]) assume static graphs with a *fixed, user-specified exposure mapping* (e.g., “fraction of treated neighbours”) and focus on bias from interference. Our setting differs in three ways:
>
> * **Dynamic graph:** the spatial grid remains fixed but covariates and treatments evolve each time step, creating lagged carry-over that standard network estimators do not usually model.
>
> * **Learned exposure function:** GST-UNet infers the spillover kernel directly from data rather than prespecifying it.
>
> * **Spatial confounding:** network methods typically ignore latent spatial fields (e.g., weather) that confound both treatment and outcome.
>
> Adapting a static-network estimator would therefore require (i) extending it to temporal graphs, (ii) specifying or learning a lagged exposure mapping, and (iii) modelling spatial confounders—all non-trivial modifications outside the scope of this paper. We believe our framework is the first to integrate all three elements within a single, consistent estimator.
>
> With respect to irregular network data, we note that the backbone is modular: swapping the grid convolution for a graph-convolution layer would let GST-UNet run on irregular networks; implementing and tuning that variant is promising future work but outside this first grid-based study.
>
> **We hope our responses have addressed your key concerns and would be grateful if you would consider a higher score in light of these clarifications.**
>
> [1] M. Tec et al. Weather2vec: Representation learning for causal inference with non-local confounding. AAAI, 2023.
>
> [2] S. Ali et al. Estimating direct and indirect causal effects of spatiotemporal interventions with spatial interference. ECML PKDD, 2024.
>
> [3] G. Ayzel et al. RainNet v1.0: A CNN for radar-based precipitation nowcasting. Geosci. Model Dev., 2020.
>
> [4] J. C. Fernández et al. Deep coastal sea forecasting using UNet-based models. Knowl.-Based Syst., 2022.
>
> [5] J. Ring et al. Prediction of sea surface temperature using U-Net. Remote Sens., 2024.
>
> [6] A. Dosovitskiy et al. An image is worth 16x16 words: Transformers for image recognition. arXiv:2010.11929, 2020.
>
> [7] K. Wu et al. TinyViT: Fast pretraining distillation for small vision transformers. ECCV, 2022.
>
> [8] M. Tec et al. SPACE: The spatial confounding environment. ICLR, 2024.
>
> [9] G. Papadogeorgou et al. Causal inference with spatio-temporal data. J. R. Stat. Soc. B, 2022.

---

> > ### Comment · Reviewer_GdrS · 2025-08-05
> >
> > Thank you for your thorough rebuttal and responses to my questions. My main questions have been addressed. As I mentioned in my initial review, I recognize the merits of the paper in filling a gap in spatio-temporal causal inference, and with the authors' rebuttal, I am now convinced about the authors' claim on the suitability of the method for data-scarce regimes, which makes it a significant advantage. While the initial submission has major presentation issues, according to the rebuttal, the authors have a clear roadmap for addressing this. I urge the authors to make sure these issues are addressed upon revision, and especially the steps under **"Re w3:"** are taken to improve the flow and clarify where the paper positions itself.
> >
> > Overall, I agree with the author's revision plans, and as far as I can tell based on this plan, I believe the revised version will resolve my main concerns. I will update my assessment and the rating accordingly.

---

> > > ### Author Response · Authors · 2025-08-05
> > > **Follow-Up and Revisions**
> > >
> > > Thank you for revisiting the paper and for the encouraging follow-up. We appreciate your acknowledgement of GST-UNet’s contribution and of our revision plan. We will make the presentation improvements you highlighted our top priority for the camera-ready. Please let us know if any additional questions arise during the discussion; we are happy to elaborate further.

---

### Author Response · Authors · 2025-08-07
**Closing Remarks**

As the discussion period is nearing its end, we thank all reviewers for their thoughtful feedback—your input has already improved and strenghtened out work.

Reviewers highlighted that **GST-UNet fills an important methodological and practical gap**: it *identifies a key gap in spatiotemporal causal representation learning and takes important steps to address it* (**GdrS**), is a *complete piece of work with clear assumptions, proofs, and theoretical background* (**XYsK, RWWK**), and is the *first architecture to unify deep spatiotemporal representation learning with iterative G-computation, handling interference, carry-over, and evolving confounders* (**sKpM**). Others noted its *practical utility in public health and environmental policy* (**zFj6**) and its thoughtful design choices, including curriculum training for G-computation heads (**GdrS**).

Building on these strengths, GST-UNet remains, to our knowledge, **the only practitioner-ready tool that provides formal identification and consistency guarantees under general time-varying confounding and spatial interference**, while being expressly designed for settings where practitioners observe just a single spatiotemporal chain—the typical case in climate, environmental-health, and mobility applications. By combining a sample-efficient ConvLSTM-UNet encoder with curriculum-stabilised iterative G-computation, GST-UNet provides **unbiased, location-specific causal-effect estimates** where existing neural or classical methods fall short.

If any further questions arise before the discussion closes, please let us know and we will respond promptly.

---

### Decision · Program_Chairs · 2025-09-17

**Decision:**

Accept (poster)

**Comment:**

The paper tackles the estimation of Conditional Average Potential Outcomes (CAPOs) from spatio-temporal observational data, facing the challenges of interference among units, spatial confounding, temporal carryover, and time-varying confounding. The difficulty and relevance of the goal w.r.t. epidemiology and earth sciences cannot be over-estimated.

The paper ambitions both to close a theory gap (extending the state of the art to time-varying confounding settings and offering  identification guarantees) and deliver a practical recipe. The rewriting of the paper will be more clearly focused on the theoretical contributions (addressing GdrS remarks).

Some reviewers (XYsK, zFj6) considered the novelty to be moderate compared to prior neural G-computation work. The authors' rebuttal clarified the difference (compared to a single spatiotemporal trajectory, the proposed approach considers each spatial location as a unit, where the learned convolutional kernel aggregates neighbour treatments/covariates before the G-comp heads), hampering the benchmarking compared to the state of the art (requiring the losses and architectures to be significantly modified). AC: perhaps the naive usage of former methods would nevertheless help to appreciate the merits and novelty of the proposed GST-Unet.

Other concerns, e.g. related to the data-scarce regime (GdrS) or the computational cost (sKpM) or the possible extension to irregular spatial graphs (sKpM) have been convincingly addressed by the authors' rebuttal.

A great suggestion from sKpM concerns the interpretability/visualisation of the influences: to be added to the revised version of the paper.